# MANF supports the inner hair cell synapse and the outer hair cell stereocilia bundle in the cochlea

Kuu Ikäheimo[1], Anni Herranen[1], Vilma Iivanainen[1], Tuuli Lankinen[1], Antti A Aarnisalo[2], Ville Sivonen[2], Kashyap A Patel[3], Korcan Demir[4], Mart Saarma[5], Maria Lindahl[5], Ulla Pirvola[1]

**Failure in the structural maintenance of the hair cell stereocilia bundle and ribbon synapse causes hearing loss. Here, we have studied how ER stress elicits hair cell pathology, using mouse models with inactivation of *Manf* (mesencephalic astrocyte-derived neurotrophic factor), encoding an ER-homeostasis-promoting protein. From hearing onset, *Manf* deficiency caused disarray of the outer hair cell stereocilia bundle and reduced cochlear sound amplification capability throughout the tonotopic axis. In high-frequency outer hair cells, the pathology ended in molecular changes in the stereocilia taper region and in strong stereocilia fusion. In high-frequency inner hair cells, *Manf* deficiency degraded ribbon synapses. The altered phenotype strongly depended on the mouse genetic background. Altogether, the failure in the ER homeostasis maintenance induced early-onset stereociliopathy and synaptopathy and accelerated the effect of genetic causes driving age-related hearing loss. Correspondingly, *MANF* mutation in a human patient induced severe sensorineural hearing loss from a young age onward. Thus, we present MANF as a novel protein and ER stress as a mechanism that regulate auditory hair cell maintenance in both mice and humans.**

## Introduction

Cochlear hair cells are the key players in hearing function. They have a compartmentalized functional organization, comprising the mechanotransduction apparatus, the hair bundle (stereocilia bundle) at their apical pole, and the synaptic contacts (ribbon synapses) at their basal pole. Of the two types of hair cells, the inner hair cells (IHCs) function as the primary receptors transducing the sound stimuli into electric signals, which are sent via the ribbon synapses to the brain. The outer hair cells (OHCs) amplify low-level sound stimuli and sharpen the frequency tuning in the cochlea (Fettiplace, 2017).

Both the hair bundles and ribbon synapses are sensitive to stressors such as loud noise and ageing. Many hereditary hearing loss syndromes, such as the Usher syndrome, also target these compartments. Perturbations in the hair bundle structure leads to the lowering of hearing sensitivity, seen as an increase in hearing thresholds. Loss of IHC ribbon synapses degrades the fidelity of sound information sent to the brain. Stressors can also trigger hair cell death, OHCs being much more vulnerable than IHCs (Fettiplace & Nam, 2019). Hair cell dysfunction and death cause sensorineural hearing loss, the most common type of hearing loss that remains without effective treatment. Therefore, a key aim of preclinical auditory research is to identify intracellular mechanisms that mediate hair cell and neuronal pathology and that could be used as targets for the development of effective treatments.

The accumulation of misfolded proteins in the ER and the concomitant $Ca^{2+}$ release from the ER stores is termed as ER stress. It is emerging as a key driver of many human diseases, including diabetes and neurodegeneration (Hetz et al, 2020). ER proteostasis imbalance activates the unfolded protein response (UPR), a signalling cascade that mediates information about the protein folding status to the cytoplasm and nucleus, with the aim to initiate an adaptive or a proapoptotic stress response (Hetz et al, 2020). MANF (mesencephalic astrocyte-derived neurotrophic factor) is an ER-resident protein that maintains normal ER homeostasis (Lindahl et al, 2014, 2017; Bell et al, 2019; Yan et al, 2019). Studies with conventional and conditional *Manf* inactivation in mice have shown that the lack of MANF triggers ER stress, best studied in the pancreatic *β* cells where *Manf* inactivation elicits chronic UPR activation and cell death (Lindahl et al, 2014, 2017). MANF is broadly expressed in mouse tissues, but only certain cell types show pathology upon *Manf* ablation (Danilova et al, 2019). *Manf* deficiency does not hamper the survival of the midbrain dopamine neurons, despite chronic UPR activation in these cells (Pakarinen et al, 2020). The mechanisms underlying the apparent context-dependent role of MANF are not well understood.

Existing evidence suggests that ER stress is detrimental to cochlear hair cells and hearing function. This was first shown by

[1]Molecular and Integrative Biosciences Research Programme, University of Helsinki, Helsinki, Finland   [2]Department of Otorhinolaryngology–Head and Neck Surgery, Helsinki University Hospital and University of Helsinki, Helsinki, Finland   [3]Institute of Biomedical and Clinical Science, College of Medicine and Health, University of Exeter, Exeter, UK   [4]Department of Paediatric Endocrinology, Dokuz Eylul University, Izmir, Turkey   [5]Institute of Biotechnology, HILIFE Unit, University of Helsinki, Helsinki, Finland

Correspondence: ulla.pirvola@helsinki.fi

exposing rats to the ER stress-inducing chemical tunicamycin (Fujinami et al, 2012). Another study showed that inactivation of the ER-localized *Tmtc4* (transmembrane and tetratricopeptide repeat 4), a regulator of ER-Ca$^{2+}$ dynamics, overactivated the UPR and caused postnatal hair cell and hearing loss (Li et al, 2018). Recently, we showed that MANF is expressed in the mouse cochlear hair cells and that *Manf* deficiency leads to robust OHC death, IHC synaptopathy and hearing loss at 8 wk of age (Herranen et al, 2020). We found that this pathology requires the C57BL/6J (B6) genetic background. The B6 background includes the *cadherin 23^{ahl}* (*Cdh23^{ahl}*) point mutation (*Cdh23^{753G→A}*), a major cause of early-onset age-related hearing loss in laboratory mouse strains (Johnson et al, 2000; Noben-Trauth et al, 2003). In another study with zebrafish hair cells, *Cdh23* mutation was shown to induce ER stress (Blanco-Sánchez et al, 2014). Based on these prior data, we hypothesized that *Manf* ablation damages mouse hair cells by potentiating the genetic background-dependent ER stress. In the present study, we provide direct evidence for this hypothesis. We show that ER proteostasis impairment triggers early-onset hair bundle structural alterations and that this dysmorphology, rather than cell death, constitute the main reason for the severe hearing loss in *Manf* mutant mice. Our mouse data together with human data on the effects of *MANF* mutation give evidence that MANF is required for proper hearing function.

## Results

### *Manf* deficiency causes hair bundle disorganization and dysfunction of OHCs from the hearing onset onward

We have previously shown that adult *Manf^{fl/fl};Pax2-Cre* cKO (conditional knock out) mice under the B6 genetic background suffer from severe hearing loss, evidenced by elevated auditory brainstem response (ABR) thresholds and robust OHC death between 5 and 11 wk of age (Herranen et al, 2020). Here, we have focused on the mechanisms behind this functional deterioration, as we hypothesized that OHC death alone cannot explain the hearing problem. To find out the age when hearing deterioration starts, we made analysis at postnatal day 15 (P15) and P22, shortly after the onset of hearing function (P12–P13). At both ages, measurements of ABRs to pure tones showed highly elevated (30–40 dB) thresholds at the frequencies of 32, 40 and 45 kHz in cKO mice as compared to age-matched control B6 mice. ABR thresholds at lower frequencies, 4–23 kHz, were also elevated, but to a lesser extent (10–20 dB). Likewise, louder stimuli were needed to generate distortion product otoacoustic emissions (DPOAEs), indicating a defect in the OHC amplifier function (Fig 1A–D). Hearing impairment became progressively worse by P56, such that the cKO mice exhibited severe to profound impairment across most of the frequency range (Fig 1E and F). Together, the hearing loss in *Manf* cKO mice has an early onset. As all OHCs and IHCs were present in mutant cochleas at the juvenile life (Fig 2A and B), OHC dysfunction, rather than OHC loss, appears to be the primary cause of the elevated hearing thresholds.

We continued to examine if OHC hair bundle morphological defects contribute to the hearing loss of *Manf* cKO mice. Whole mount specimens were labelled with phalloidin that probes for actin filaments (F-actin), the major structural component of the stereocilia. Scanning electron microscopy (SEM) was used in parallel. Phalloidin labeling showed that OHC hair bundles of cKO mice were disorganized at P15 and P22 (Fig 2C and D). SEM analysis revealed that this disorganization involved stereocilia splaying, but the numbers and length of stereocilia appeared unaltered. Both high-frequency (45 kHz, Fig 2E–H) and low-frequency (16 kHz, Fig S1A–F) OHCs showed abnormalities, which looked like the bundle cohesion was impaired. In contrast to OHCs, the morphology of IHC hair bundles appeared comparable in mutant and control mice (Fig 2I and J). These findings suggest that the OHC functional impairment, detected by DPOAEs at the juvenile stages, was due to hair bundle dysmorphology. To find out if *Manf* inactivation has an effect on the hair bundles already before the onset of hearing, we prepared whole mounts for SEM analysis at P9. We could not see obvious differences in hair bundle morphology between cKO and control mice (Figs 3A–D and S2A–D). Also, we found an array of inter-stereociliary links (Richardson & Petit, 2019) in the hair bundles of both genotypes, suggesting that bundle cohesion was unaffected at P9 (Figs 3E and F and S2E and F). However, even though our analysis did not reveal morphological defects in the immature OHC hair bundles in mutant mice, we cannot exclude the possibility of more fine-grained changes in their stereocilia.

### *Manf* deficiency leads to progressive fusion of the OHC stereocilia under C57BL/6J background

The OHC hair bundle disorganization in juvenile *Manf* cKO mice progressed to more severe dysmorphology, displayed in phalloidin labeling as fragmented bundles at P56 (Fig 4A–D). Based on SEM analysis at this age, high-frequency OHCs (45 kHz) showed strong fusion of stereocilia, whereas low-frequency OHCs (16 kHz) exhibited milder disorganization, consistent with the ABR and DPOAE data (Fig 1E and F). The first signs of stereocilia fusion were observed at P35 when the edges of hair bundles of several 45-kHz OHCs were fused (Fig 4I and J). By P56, the pathology progressed to the formation of distinct apical projections comprising groups of stereocilia enveloped by the plasma membrane (Fig 4K). As evidenced by the imprints of stereocilia left on the tectorial membrane, stereocilia had been in contact with the tectorial membrane at some point in time, but then detached before fusion (Fig 4L). Analysis by transmission electron microscopy (TEM) displayed lifting of the OHC apical membrane and confirmed that the clumped stereocilia had lost their individuality because of plasma membrane fusion. The F-actin-packed rootlets extend from the cuticular plate toward stereocilia tips, normally one rootlet per stereocilium (Pacentine et al, 2020). In mutant OHCs, each abnormal apical projection harboured several rootlets, likely collected from the group of stereocilia that had fused together (Fig 4M–O). Fused stereocilia cannot bend relative to one another and, hence, it is likely that no tension is generated for the tip links to function in mechanoelectrical transducer channel opening.

As most OHCs with fused stereocilia lacked signs of cell body degeneration, such as cell shrinkage and nuclear deformation, the stereocilia fusion appeared not to be just a by-product of ongoing cell death (Fig 4P). However, the eventual fate of these cells is likely

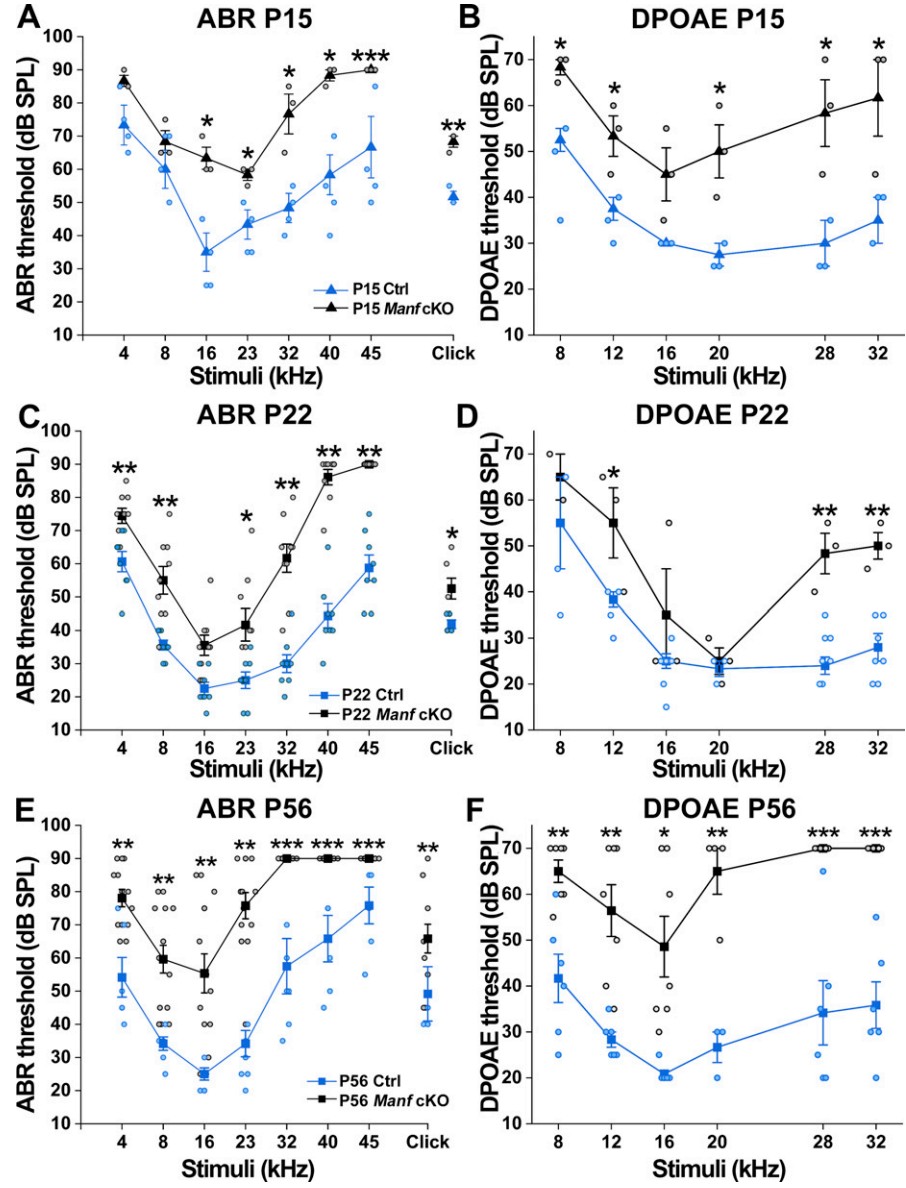

**Figure 1. Progressive deterioration of hearing function in *Manf*^(fl/fl);*Pax2-Cre* B6 mice.**

**(A)** Compared to controls, cKO mice exhibit elevated auditory brainstem response (ABR) thresholds already at P15. Circles represent measurements from individual animals. * = $P < 0.05$, ** = $P \leq 0.001$. At 45 kHz, none of cKOs show detectable ABRs (***). One-way ANOVA $P$-values: tone stimuli 4 kHz $P = 0.057$, 8 kHz $P = 0.398$, 16 kHz $P = 0.005$, 23 kHz $P = 0.014$, 32 kHz $P = 0.006$, 40 kHz $P = 0.006$; click stimulus: $P = 0.039 \times 10^{-2}$. Animals (cochleas) per group: cKO $n = 3$, control $n = 4$. Error bars show ± SE. **(B)** At P15, cKO mice also show higher DPOAE thresholds than controls (y-axis = L2, x-axis = geometric mean frequency of f1 and f2, see the Materials and Methods section). One-way ANOVA $P$-values: 8 kHz $P = 0.025$, 12 kHz $P = 0.025$, 16 kHz $P = 0.060$, 20 kHz $P = 0.031$, 28 kHz $P = 0.020$. 32 kHz $P = 0.050$. Animals (cochleas) per group: cKO $n = 3$, control $n = 3$. **(C)** At P22, ABR thresholds are higher in cKOs than in controls throughout the frequency range, most prominently in high frequencies. The higher thresholds of P15 than P22 mice is likely caused by the immaturity of the P15 auditory system. One-way ANOVA $P$-values: tone stimuli 4 kHz $P = 0.002$, 8 kHz $P = 0.001$, 16 kHz $P = 0.544$, 23 kHz $P = 0.010$, 32 kHz $P = 0.002 \times 10^{-5}$, 40 kHz $P = 0.005 \times 10^{-5}$, 45 kHz $P = 0.003 \times 10^{-5}$; click stimulus: $P = 0.007$. Animals (cochleas) per group: cKO $n = 9$, control $n = 8$. **(D)** At P22, DPOAE thresholds are higher over much of the frequency range of cKO mice than controls. One-way ANOVA $P$-values: 8 kHz $P = 0.258$, 12 kHz $P = 0.020$, 16 kHz $P = 0.122$, 20 kHz $P = 0.643$, 28 kHz $P = 0.001$, 32 kHz $P = 0.001$. Animals (cochleas) per group: cKO $n = 3$, control $n = 7$. **(E)** At P56, cKO mice exhibit highly elevated ABR thresholds at all frequencies. At 32 kHz and higher, their ABRs could not be detected using the 90 dB stimuli. One-way ANOVA $P$-values: tone stimuli 4 kHz $P = 0.045 \times 10^{-2}$, 8 kHz $P = 0.039 \times 10^{-2}$, 16 kHz $P = 0.001$, 23 kHz $P = 0.0014 \times 10^{-5}$; click stimulus $P = 0.001$. Animals (cochleas) per group: cKO $n = 14$, control $n = 8$. Error bars show ± SE. **(F)** At P56, no DPOAEs could be measured above 20 kHz using the L2 at 70 dB and only one cKO mouse produced a measurable response at 20 kHz. One-way ANOVA $P$-values: 8 kHz $P = 0.001$, 12 kHz $P = 0.001$, 16 kHz $P = 0.003$, 20 kHz $P = 0.001$. Animals (cochleas) per group: cKO $n = 7$, control $n = 6$. Abbreviations: ABR, auditory brainstem response; B6, C57BL/6J strain; cKO, conditional knock out; Ctrl, control; DPOAEs, distortion product otoacoustic emissions; OHC, outer hair cell; SPL, sound pressure level.

to be death, evidenced by the high numbers of lost OHCs in the high-frequency region of *Manf* mutant cochleas at P56 (Fig 4G and H). Unlike OHCs, IHCs showed a normal hair bundle morphology and their survival was not abrogated (Fig 4Q and R). This was despite the fact that MANF is normally expressed in both hair cell types (Herranen et al, 2020).

### *Manf* deficiency leads to molecular changes in the cuticular plate and stereocilia base of OHCs under C57BL/6J background

To reveal how the stereocilia fusion is formed in OHCs of *Manf* cKO mice, we focused on the taper region, the region at the base of stereocilia that provides structural stability to the hair bundle (Salles et al, 2014). The taper region harbours a multi-protein complex that includes PTPRQ (protein tyrosine phosphatase receptor Q), radixin and myosin 6. PTPRQ is a plasma membrane lipid

phosphatase, radixin is a member of the ezrin-radixin-moesin family that cross-links the actin filaments to the plasma membrane, and myosin 6 is an actin-associated motor protein (Hasson et al, 1997; Goodyear et al, 2003; Pataky et al, 2004). In loss-of-function mouse models, depletion of any one of the taper proteins leads to stereocilia fusion with mislocalization of the other proteins toward stereocilia tips (Self et al, 1999; Goodyear et al, 2003; Kitajiri et al, 2004; Sakaguchi et al, 2008; Salles et al, 2014; Seki et al, 2017). We found in OHCs of P56 control mice that PTPRQ expression was undetectable, myosin 6 was strongly expressed in the taper region and the cuticular plate, and radixin was enriched in the taper region. We found PTPRQ up-regulation in the taper region of OHCs of mutant mice. Furthermore, PTPRQ, radixin, and myosin 6 were abnormally localized toward the tips of fused stereocilia (Fig 5A–L′). In contrast, IHCs of mutant specimens lacked changes in the expression of taper proteins, consistent with the unaltered IHC hair

**Figure 2.  Juvenile *Manf^fl/fl^;Pax2-Cre* B6 mice show hair bundle disorganization.**
**(A, B)** At P22, there is no outer hair cell (OHC) or inner hair cell (IHC) death in the cochleas of cKO mice, shown in the 45-kHz (high frequency) region by myosin 7a immunostaining. The three rows are OHCs are marked by numbers. **(C, D)** Comparison of phalloidin labeling in the mutant and control cochleas at P22 reveals the OHC hair bundle disorganization in the 45 kHz region of mutant specimens. Arrowheads point to hair bundle imperfections. **(E, F)** SEM analysis at P22 confirms the hair bundle disorganization (arrows) in mutant specimens. **(G, H)** High-magnification views of 45-kHz OHC hair bundles display problems in individual stereocilia to hold their upright position in the bundle in mutant specimens (arrow). **(I, J)** High-magnification views of 45-kHz IHC hair bundles show comparable organization between the genotypes. Abbreviations: B6, C57BL/6J strain; cKO, conditional knock out; Ctrl, control; IHCs, inner hair cells; OHC, outer hair cell; SEM, scanning electron microscopy. Scale bars: (A) 5 $\mu$m A, B; (C) 5 $\mu$m C, D; (E) 5 $\mu$m E, F; (G) 1 $\mu$m G, H; (I) 1 $\mu$m I, J.

bundle morphology (Fig S3A–L'). These results coupled with TEM analysis (Fig 4M–O) suggest that sustained ER stress leads to re-organization of the OHC taper region proteins, leading to distur-bances in the normal actin-plasma membrane linkages and to stereocilia fusion.

We next studied if the up-regulation and mislocalization of the taper proteins are linked to the progression of OHC hair bundle pathology. *Manf* cKO mice at P22 displayed disorganized bundles but no stereocilia fusion (Fig 2D–H). At this age, we did not detect PTPRQ up-regulation in the taper region or mislocalization of PTPRQ or radixin toward the tips of stereocilia (Fig S3M–P'). At P56,

when stereocilia of 45-kHz OHCs were strongly fused, 16-kHz OHC hair bundles showed modest disorganization (Fig 4A, B, and F) and occasionally PTPRQ up-regulation in the taper region (Fig S3Q–R'). These findings reflect the progression of the hair bundle pathology and suggest that the molecular changes in the taper region are closely linked to the stereocilia fusion event.

In OHCs, the ER is accumulated to the region just beneath the cuticular plate (Mammano et al, 1999). The F-actin-based cuticular plate serves as an anchoring site for stereocilia (Tilney et al, 1980; Pacentine et al, 2020). We found strong phalloidin labeling in the OHC cuticular plates in the 16-kHz frequency region of both the

**Figure 3. Pre-hearing *Manf^{fl/fl};Pax2-Cre* B6 mice show normal morphology of the outer hair cell (OHC) hair bundles.**
**(A, B)** At P9, SEM analysis in the 45-kHz cochlear region in cKO and control mice showed a comparable appearance of OHC hair bundles. Hair bundle orientation and cohesion appeared unperturbed by the mutation at this immature age. **(C, D)** High-magnification images of single bundles show no apparent differences in stereocilia length or organization. **(E, F)** Close-up images of 45-kHz OHC hair bundles show the presence of inter-sterociliary links (arrowheads) in both genotypes. Abbreviations: B6, C57BL/6J strain; cKO, conditional knock out; OHC, outer hair cell; SEM, scanning electron microscopy. Scale bars: (A) 5 µm A-B; (C) 1 µm C-D; (E) 200 nm E-F.

control and *Manf* cKO cochleas at P56 (Fig 5M and N). In contrast, in the 45-kHz region, the cuticular plates of mutants showed strongly reduced phalloidin labeling, concomitantly with fused stereocilia (Fig 5O–P'). Quantification of phalloidin fluorescence yielded a significant difference between mutant and control specimens. This difference was not found when 16-kHz OHCs were compared (Fig 5Q and R). Corresponding changes were not seen in the expression of non-erythroid α-spectrin, another cytoskeletal component of the cuticular plate (Ylikoski et al, 1992) (Fig S3S and T). Unlike OHCs, IHCs did not show a difference in cuticular plate phalloidin fluorescence between the genotypes at P56 (Fig 5S and T). TEM analysis supported these F-actin data by showing lower electron density in the cuticular plates of mutant OHCs with fused stereocilia compared to controls (Fig 5U and V). Together, the cuticular plate appears to lose its normal F-actin network concomitantly with stereocilia fusion and with changes in the expression of the taper region proteins.

MANF was recently found to physically interact with neuroplastin, an immunoglobulin superfamily glycoprotein (Yagi et al, 2020). Interestingly, neuroplastin can form a functional complex with the plasma membrane $Ca^{2+}$ ATPases (PMCAs) in the ER, and this interaction is critical for the stability and trafficking of PMCAs to the plasma membrane (Schmidt et al, 2017). Particularly interesting for the current study is that both neuroplastin and PMCA2 are strongly expressed in the OHC stereocilia and are critical for hearing function (Zeng et al, 2016; Fettiplace & Nam, 2019). Furthermore, neuroplastin has been shown to be required for PMCA2 expression in the OHC stereocilia (Lin et al, 2021). Because of this potential link between MANF and the maintenance of $Ca^{2+}$ homeostasis in OHC

stereocilia, we next examined neuroplastin and PMCA2 expression in OHCs of *Manf* cKO and control mice. We did not detect differences between the genotypes at the juvenile age (P15, Fig S4A–B"). We did detect decreased expression of both proteins in the high-frequency OHCs of cKO mice at young adulthood (P35) when stereocilia fusion was starting (Fig 6A–B" and 4J). In the adults, neuroplastin and PMCA2 down-regulation correlated with the extent of stereocilia fusion: low-frequency OHCs lacked changes, whereas in the high-frequency cochlear region adjacent OHCs displayed a stark contrast in the expression levels (Figs 6C and C" and S4C–D"). Thus, the decrease in neuroplastin and PMCA2 did not match with the functional deficit at the juvenile age, but matched with the OHC stereocilia fusion and robust functional impairment at adulthood.

### The susceptibility of OHCs to *Manf* deficiency is dependent on the genetic background

Our prior findings (Herranen et al, 2020) suggested that the effect of *Manf* inactivation depends on the underlying mouse strain, such that OHC loss and ABR threshold elevations were manifested in *Manf* KO mice only under the genetic background that carries the $Cdh23^{753G\rightarrow A}$ mutation (B6 and CD-1 backgrounds, Johnson et al, 2000; Noben-Trauth et al, 2003). *Manf* inactivation under the CBA/Ca (CBA) background, lacking this mutation (Noben-Trauth et al, 2003), showed a well-preserved OHC population and no ABR threshold elevations (Herranen et al, 2020). The next important question was whether also the OHC hair bundle dysmorphology was dependent on the combined effect of *Manf* inactivation and

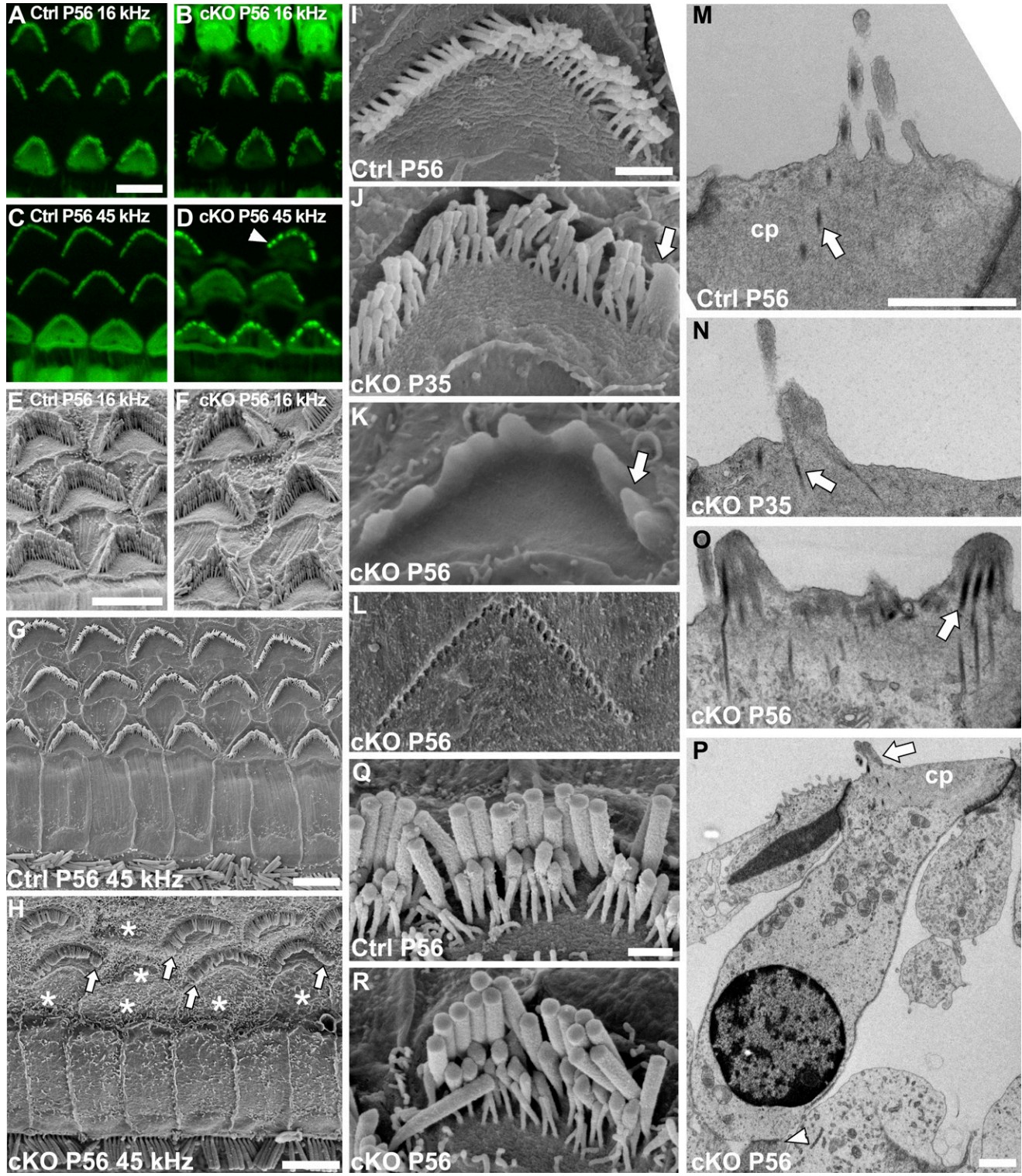

**Figure 4. Adult Manf^fl/fl;Pax2-Cre B6 mice show progressive deterioration of the outer hair cell (OHC) hair bundles.**
**(A, B)** At P56, comparison of phalloidin labeling in the 16-kHz region of mutant and control cochleas shows disorganized OHC hair bundles only in mutants **(C, D)** Phalloidin labeling in the 45-kHz region of mutant specimens shows severe bundle disorganization, in the form of bundle fragmentation (arrowhead). **(E, F)** Scanning electron microscopy (SEM) images from the 16-kHz region confirms the loss of cohesion of OHC bundles in mutant mice. **(G, H)** SEM images from the 45-kHz cochlear region show OHCs deterioration in cKO mice, such that most OHCs have fused stereocilia (arrows) and many of them are lost (*, site of lost OHC). **(I, J, K)** SEM views showing age-related progression in hair bundle deterioration in cKO mice. The bundles exhibit stereocilia fusion at their edges (arrow) at first, shown at P35. The fusion progresses and affects the entire bundle, shown at P56. **(L)** Stereocilia imprints of a high-frequency OHC embedded in the tectorial membrane in P56 mutant specimen are shown (see text for details). **(M, N, O)** Transverse transmission electron microscopy (TEM) sections of high-frequency OHCs from mutant and control mice. The sections from P35 and P56

strain-specific background. To examine this, we bred the original *Manf* KO CD-1 mouse line (Lindahl et al, 2014) toward pure CBA background for five to six additional generations. We confirmed by sequencing that the mice generated lacked the $Cdh23^{753G \to A}$ mutant allele, in heterozygous or homozygous form. We found that the *Manf* KO CBA mice had normal ABRs and they had OHCs with a normal appearance of hair bundles, based on SEM analysis at P56 (Fig 7A–C). As expected from the lack of stereocilia fusion, PTPRQ was not expressed in the taper region in their OHCs (Fig 7D and D'). Thus, *Manf* inactivation required the concomitant genetic predisposition to induce the degradation of the OHC mechanotransduction organelle and hearing loss (Table 1).

### *Manf* deficiency leads to synaptopathy and abnormalities in synapse morphology in the IHCs of C57BL/6J mice

We have previously shown that ~50% of ribbon synapses are lost in the high-frequency IHCs of *Manf* cKO mice at P56 (Herranen et al, 2020). Despite this synaptopathy, IHCs were otherwise unaffected, evidenced by the lack of signs of cell body degeneration and hair bundle dysmorphology (Fig 4Q and R). We now wished to understand when and how IHC synaptopathy takes place in mutant mice. We used CtBP2 immunostaining to label presynaptic ribbons. Quantification of CtBP2-positive ribbons per IHC in the 45-kHz region revealed an equal number of these structures in cKO and control mice at P22, whereas 50% of them were lost in cKOs already at P35 (Fig 8A–E). The ribbon loss at P35 was similar to the loss quantified at P56 (Herranen et al, 2020). The mean size of the remaining ribbons at P35 was also larger in cKO cochleas (Fig 8F). These results demonstrate that IHC synaptopathy in the high-frequency region of mutant cochleas takes place at a restricted time period at young adulthood. Interestingly, in contrasts to IHCs, OHCs of mutant mice lacked apparent alterations in ribbon synapses, based on CtBP2 immunofluorescence that showed the normal amount of ribbons (one-to-three) per OHC (Fig S5A and B).

Quantification of the position of synaptic ribbons across the IHC modiolar-pillar axis did not yield a statistically significant difference between *Manf* cKO and control mice at P56 (Fig 8G and H). Thus, in this mutant mouse model with ER stress, synaptopathy appears not to involve synapses of a particular position on the modiolar-pillar axis and thereby it cannot be linked with a specific subtype of ribbon synapses (Taberner & Liberman, 2005; Ohn et al, 2016).

To examine if the remaining presynaptic ribbons were juxtaposed by the postsynaptic receptor patch, we used immunostaining for the postsynaptic density-localized scaffolding protein Homer1 (Martinez-Monedero et al, 2016). There was almost perfect pairing of the pre- and postsynaptic receptor components in control specimens; only a few "orphan" ribbons were found. This was the case with the non-synaptopathic mutant cochleas at P22 as well (Fig 8I

and J). In contrast, in the 45-kHz region of mutant cochleas at P35 (Fig 8K and L) and P56, there were not only fewer presynaptic ribbons than normal, but a significant proportion of the remaining ribbons were "orphans" (P35 control mean ± SD 2.0% ± 1.5%, n = 3 mice; P35 cKO 23.4% ± 8.8%, n = 2; P56 control 4.9% ± 2.7% n = 3; P56 cKO 18.7% ± 5.1% n = 3, one cochlea per mouse, 20 IHCs per cochlea, binomial test: P35 *P* < 0.001, P56 *P* < 0.001). Correspondingly, the quantity of Homer1-positive postsynaptic receptor components in the 45-kHz region was lower in the P35 mutant cochleas as compared with age-matched controls (P35 control mean 15.9 ± 2.8, 95% CI [13.1, 18.2]; P35 cKO 8.2 ± 2.5, 95% CI [5.5, 10.6]; estimate of difference 7.6, *P* = 0.004; linear mixed model, type 3 test of fixed effect, n = 3 mice both groups). These findings demonstrate that the ribbon synapse damage involved both the pre- and postsynaptic structures and indicate severe synaptic pathology.

Most of the CtBP2 puncta appeared small-sized in the IHCs of control mice at P22, P35, and P56. This was also the case with *Manf* cKO mice at P22. However, in older mutants with synaptopathic IHCs, many of the remaining CtBP2 immunofluorescent puncta appeared abnormally large (Fig 8D and F). A possibility was that this is due to two very closely located normal-sized ribbons, which is difficult to confirm by immunofluorescence detection. Therefore, we continued with TEM analysis. A small portion of the IHC ribbon synapses in the normal adult cochlea harbours two ribbons at the same active site (Sobkowicz et al, 1982; Michanski et al, 2019). Double ribbon synapses were found both in our control and mutant specimens (Fig 8M–P). These latter specimens showed large variability in the proportion of double ribbons and the results did not allow us to conclude that the large-sized CtBP2 fluorescent puncta reflect increased number of double ribbons. TEM sections revealed that several ribbons in mutant specimens were abnormally shaped and were not anchored to the active zone. Yet, a normal-appearing halo of synaptic vesicles was tethered to the ribbons, even in the rare cases in which ribbons were "free-floating" in the cytoplasm far away from the active site (Fig 8Q). Despite these alterations in synapse organization, the terminal boutons of the afferent spiral ganglion neurons lacked swelling or membrane ruptures (Fig 8R and S), suggesting that glutamate excitotoxicity (Puel et al, 1998) is not a driving force for the ribbon synapse dysmorphology in *Manf* cKO mice.

Otoferlin is a $Ca^{2+}$ sensor specific for the IHC ribbon synapses. It is required for the exocytosis of synaptic vesicles (Roux et al, 2006). Prior data have demonstrated synaptopathy in *otoferlin*-inactivated IHCs (Vincent et al, 2017; Tertrais et al, 2019). These data together with the knowledge that otoferlin is a member of the TA (tail-anchored)-family proteins that are inserted to the ER via the TRC40 receptor WRB (Lin et al, 2016; Vogl et al, 2016) made us hypothesize that synaptopathy in *Manf* cKO mice is associated with altered otoferlin expression. However, we did not find differences in otoferlin expression between mutant and control mice at any age

---

mutant specimens confirm stereocilia fusion and show apical membrane lifting that creates this fusion. The rootlets (arrows) remain separate in the fused structures. **(P)** A transverse TEM section of a 45-kHz OHC from a P56 mutant specimen shows no signs of cell pathology other than the hair bundle fusion. Both efferent (arrowhead) and afferent (not shown) synaptic connections can be found. **(Q, R)** SEM images reveal a comparable appearance of 45-kHz IHC hair bundles in P56 mutant and control mice. Abbreviations: B6, C57BL/6J strain; cKO, conditional knock out; cp, cuticular plate; Ctrl, control; IHC, inner hair cell; OHC, outer hair cell; SEM, scanning electron microscopy; TEM, transmission electron microscopy. Scale bars: (A) 5 μm A-D; (E) 5 μm E, F; (G) 5 μm; (H) 5 μm; (I) 1 μm I-L; (M) 1 μm M-O; (P) 1 μm; (Q) 1 μm Q, R.

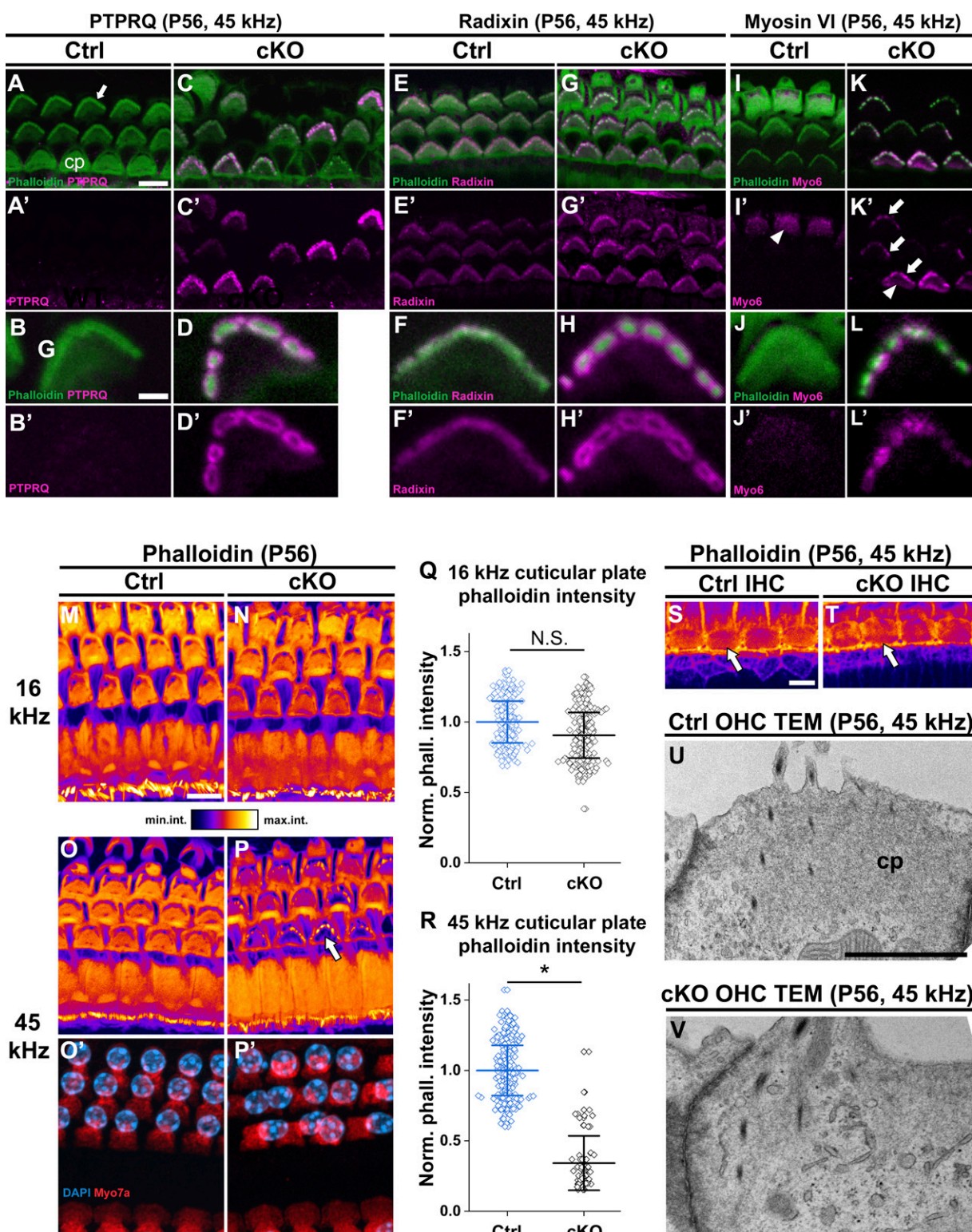

**Figure 5. Stereocilia fusion is associated with molecular and morphological changes in the outer hair cells (OHCs) in *Manf^{fl/fl};Pax2-Cre* B6 mice.**
**(A, A', B, B', C, C', D, D')** At P56, PTPRQ immunostaining coupled with phalloidin labeling shows that PTPRQ expression is undetectable in the 45-kHz OHCs of control mice, whereas comparable OHCs of cKO mice show PTPRQ up-regulation in the fused stereocilia and the taper region. PTPRQ envelopes the F-actin core of the fused stereocilia. **(E, E', F, F', G, G', H, H')** At P56, radixin is weakly expressed in the OHC taper region in control mice, whereas it is mislocalized up to the fused stereocilia in OHCs of mutant mice. Also radixin envelopes the F-actin core of the fused stereocilia. **(I, I', J, J', K, L, L')** At P56, myosin 6 is expressed in the OHC cuticular plate (arrowheads) and cell body in both genotypes. Similar to PTPRQ and radixin, myosin 6 expression is detected in the fused stereocilia (arrows). The oblique section shows only some OHCs at the stereocilia level, thus leaving out the signal from the cuticular plate but illustrating the stereociliary myosin 6 in the mutant. **(M, N, O, O', P, P')** Phalloidin

studied (Fig 8T and U), suggesting that ER stress-induced synaptopathy is regulated by other mechanisms.

Finally, as our results showed that *Manf* ablation leads to OHC hair bundle pathology under B6 but not under CBA background (Fig 7C and Table 1), we investigated whether this was the case with the IHC ribbon synapse defects as well. Quantification at P56 showed comparable numbers of CtBP2 puncta in IHCs of *Manf* KO CBA mice and control littermates (Fig S6A–C), this being in stark contrast to the 50% ribbon loss seen in IHCs of age-matched *Manf* cKO B6 mice.

### *MANF* mutant patient exhibits early and severe sensorineural hearing loss

Recently, recessive loss-of-function variants in *MANF* have been shown to cause childhood-onset diabetes, sensorineural hearing loss, microcephaly, and developmental delay (Yavarna et al, 2015; Montaser et al, 2021). We have here characterized the audiological phenotype of case 1 from the study by Montaser et al (2021). Detailed audiological data from case 2 was not available. Our patient case is a female. No newborn hearing screening was performed. One incidence of otitis media was diagnosed during early childhood. Other possible mutations contributing to the early hearing phenotype and familial incidences of genetic hearing loss are unknown. Hearing was evaluated for the first time at 11 mo of age. Click-evoked ABR testing was performed. Mature and symmetric waves I – V could be seen. At 70 dBHL stimulus level V-wave could not be identified at both sides (Fig 9A and B). Concurrently, auditory steady state responses were measured. At ~500 Hz frequency, 65 dB threshold was seen on both sides. These tests suggest a severe hearing impairment (WHO's grades of hearing impairment).

Clinical standardized audiometry was performed at 10 yr of age. Pure tone averages were 83/73 dBHL (500–4,000 Hz) (Fig 9C). Bone conduction averages were 64 (masked)/54 (unmasked) dBHL (500–4,000 Hz). Tympanometry showed sharp maximum compliance peaks with slight negative pressure to normal middle ear pressure (−205 d Pa/−125 d Pa). Clinical standardized speech audiometry was performed with 85/80 dBHL unaided speech recognition thresholds. Unaided discrimination speech scores were 68%/76%. At 14 yr of age, clinical audiometry showed symmetric sensorineural hearing loss with pure tone average 60/65 dBHL (Fig 9D). Audiometric testing shows severe symmetric sensorineural hearing loss, similar to *Manf* cKO mice. Also similar to our mutant mice, the patient lacked balance defects.

## Discussion

Here, we provide evidence of a relationship between age-related hearing loss and elevated ER stress susceptibility. We have studied mouse strains in which the ER-resident, chaperone-like MANF is depleted from the cochlea. The strain predisposed to age-related hearing loss (B6) responded strongly to *Manf* inactivation, in contrast to the strain (CBA) lacking this predisposition. Our results show that the depletion of a component of the ER-homeostasis-regulating machinery exacerbates the effects of the genetic background, the combined effect causing early-onset problems in the maintenance of the key functional domains of hair cells, the OHC hair bundle and the IHC ribbon synapse.

Age-related hearing loss is typical to several commonly used mouse strains, the hypomorphic *Cdh23* allele $Cdh23^{753G\to A}$ (known as $Cdh23^{ahl}$) being a major cause (Johnson et al, 2000; Noben-Trauth et al, 2003; Ohlemiller et al, 2016). CDH23 is a component of the stereocilia tip links, required for gating the mechanotransduction channels (Siemens et al, 2002; Söllner et al, 2004). B6 mice carry the $Cdh23^{753G\to A}$ mutation that causes defects in the hair bundle maintenance and hair cell survival at adulthood. Studies with the zebrafish hair cells have shown that *Cdh23* mutation impairs the trafficking of CDH23 from the ER to the Golgi compartment and causes ER stress. This was shown to lead to hair bundle defects and hair cell death (Blanco-Sanchez et al, 2014). Comparable cell biological disturbances might underlie the hair cell pathology in B6 mice, yet no direct evidence exists. When we studied B6 mice before their normal hearing decline and included *Manf* inactivation, we saw high numbers of dysmorphic OHC hair bundles together with elevated ABR and DPOAE thresholds. This suggests that the combination of the $Cdh23^{753G\to A}$ mutation and *Manf* inactivation exceeds the critical threshold of ER stress above which OHC pathology ensues, leading to acceleration of age-related hearing loss. The capacity of the ER proteostasis network, including MANF as one of its components, has been shown to decline with ageing in tissues of both animals and humans (Neves et al, 2016; Taylor & Hetz, 2020). This could be the case in the cochlear hair cells as well, and the down-regulation of MANF or components of the UPR could have an exacerbating effect on age-related progressive hearing loss.

In *Manf* cKO mice, OHC hair bundle abnormalities (this study) well-preceded the death of these cells (Herranen et al, 2020). Likewise, *Manf* inactivation triggered IHC synaptopathy, but IHC survival was unaffected. Most studies addressing the

fluorescence illustrated with an intensity-coded lookup table in the 16 and 45-kHz OHCs of mutant and control mice. Images are maximum intensity projections from equal-size image stacks encompassing stereocilia and the cuticular plate. Phalloidin fluorescence in the cuticular plate of 16-kHz OHCs is comparable in the two genotypes (M and N). In contrast, the fluorescence is strongly reduced (arrow) in 45-kHz OHCs in mutant mice (O and P). **(O', P')** Myosin 7a immunostaining (red) and DAPI counterstain (blue) show comparable cell and nuclear morphologies, excluding the possibility that the F-actin abnormalities are a sign of cell degeneration. **(Q, R, S, T)** Quantification of the phalloidin fluorescence intensity data: Normalized mean intensity ± SD of fluorescence in the cuticular plates of 16-kHz (S) and 45-kHz (T) OHCs in mutant and control mice are shown as horizontal bars with whiskers. Data points represent normalized phalloidin intensity values in the cuticular plate of single OHCs. In the 16 kHz region, the mean phalloidin intensity is not significantly different between the genotypes. The difference is statistically significant in the 45 kHz region, mutants showing strongly reduced phalloidin compared to controls. 16 kHz $P = 0.507$; 45 kHz $P = 0.045$, linear mixed model, type 3 test of fixed effect (genotype). Animals (cochleas) per group: control $n = 3$, cKO $n = 3$. *$P < 0.05$; N.S, not significant. **(S, T)** Phalloidin fluorescence comparison shown in IHC cuticular plates (oval sectors, arrows) in the 45-kHz region of mutant and control mice. **(U, V)** TEM sections through the apical part of 45-kHz OHCs in control specimen show a strongly electron-dense cuticular plate in control specimen, whereas this structure has a weak appearance in mutant specimen. Note also OHC stereocilia fusion in the mutant specimen. Abbreviations: B6, C57BL/6J strain; cKO, conditional knock out; cp, cuticular plate; Ctrl, control; IHC, inner hair cell; Myo6, myosin 6; OHC, outer hair cell; PTPRQ, Protein Tyrosine Phosphatase Receptor Type Q; TEM, transmission electron microscopy. Scale bars: (A) 5 μm A, C, E, G, I, K; (B) 1 μm B, D, F, H, J, L; (M) 5 μm M-P'; (S) 5 μm S, T; (U) 1 μm U, V.

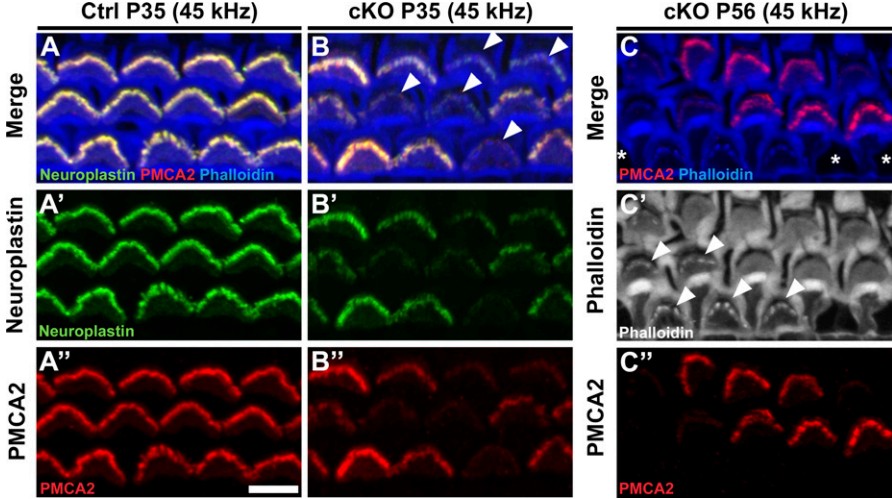

**Figure 6. Neuroplastin and PMCA2 expressions are down-regulated in the outer hair cells (OHCs) with fused stereocilia in *Manf^{fl/fl};Pax2-Cre* B6 mice.**
**(A, B)** At P35, double-immunostaining for neuroplastin and PMCA2, coupled with phalloidin labeling, shows co-localization of these proteins in the 45-kHz OHC hair bundles in both control and cKO mice. In mutant specimen, many bundles show weak neuroplastin and PMCA2 expression (arrowhead). **(A′, A″, B′, B″)** The stainings are separately shown in these views. The down-regulation of both expressions is seen in the mutant specimen. **(C, C″)** At P56, PMCA2 expression is weak or undetectable in those OHCs that exhibit strong stereocilia fusion, as interpreted from phalloidin labeling (arrowheads in C′). Asterisks mark sites of lost OHCs. Abbreviations: B6, C57BL/6J strain; cKO, conditional knock out; Ctrl, control; OHC, outer hair cell; PMCA2, plasma membrane Ca²⁺ ATPase 2. Scale-bar: (A″) 5 µm in all images.

pathophysiological role of ER stress have focused on the cell death programmes. Our results here suggest that ER stress impacts on the maintenance of the critical functional domains of hair cells separately from its effect on cell death induction. This is important considering the progressive nature of age-related hearing loss: at least in the mouse, hair cell structural defects constitute a major underlying cause of permanent hearing loss, and hair cell death takes place at later stages (Kujawa & Liberman, 2019). However, as we found in the cochleas of *Manf* cKO mice that the final stage of OHC hair bundle pathology—stereocilia fusion—coincided temporally and regionally with OHC death, it is possible that these two events are interrelated and happen in sequence, by yet poorly understood mechanisms.

*Manf* cKO B6 mice displayed OHC hair bundle disarray already at juvenile ages (P15 and P22). The bundle disarray was subtle at this stage that represents the onset of hearing function. However, DPOAE and ABR thresholds were already prominently raised. Even though we did not see bundle abnormalities at the immature stage (P9), we cannot exclude possible fine-grained developmental abnormalities that could contribute to the hearing impairment. The mechanisms are elusive. Already in immature OHCs, *Manf* inactivation could perturb the production or trafficking of stereociliary proteins, particularly the mutant CDH23 protein (B6 background). These defects could affect the structure and function of tip links and transient lateral links, manifested at the onset of hearing as problems in hair bundle cohesion and cell dysfunction. The OHC pathology in *Manf* cKO mice progressed thereafter. Chronic ER stress in adult OHCs due to *Manf* inactivation might exacerbate the effect of *Cdh23^{753G→A}* mutation, known to drive age-related hearing loss in B6 mice, and this combination might cause earlier onset of hearing impairment. However, giving definitive proof for this concept in the cochlea in vivo requires novel methods to measure ER stress dynamics in hair cells. In addition, it would be important to show that *Cdh23* missense mutation indeed elicits ER stress in the mouse cochlear hair cells similarly as shown for *Cdh23* mutation in the neuromast hair cells of the *sputnik* zebrafish model (Blanco-Sanchez et al, 2014).

The end result of OHC hair bundle pathology, stereocilia fusion, was unusually prominent in our mutant mouse model. How could ER stress regulate stereocilia fusion in OHCs? The taper region at the base of stereocilia comprises a multi-protein complex that stabilizes the linkages between the plasma membrane and actin filaments. Depletion of taper proteins has been shown to cause stereocilia fusion (Self et al, 1999; Goodyear et al, 2003; Kitajiri et al, 2004; Sakaguchi et al, 2008; Salles et al, 2014; Seki et al, 2017). We found that OHC stereocilia fusion was associated with changes in the expression of taper proteins. Radixin and myosin 6 were mislocalized from the taper region upward along the fused stereocilia. Most interestingly, the plasma membrane lipid phosphatase PTPRQ, which was absent from OHCs in control adult mice, was strongly up-regulated both in the taper region and in the more distal parts of the fused stereocilia. As a phosphatidylinositol phosphatase, PTPRQ has been suggested to regulate the compartmentalized distribution of phosphatidylinositol 4,5-bisphosphate (PIP₂) in the stereocilia plasma membrane. Thereby, PTPRQ could be involved in the actin-plasma membrane dynamics (Hirono et al, 2004; Richardson & Petit, 2019). Together, ER stress leads to changes in the expression of taper proteins and membrane lipids and to modification of actin-plasma membrane linkages, leading to stereocilia fusion in OHCs of *Manf* cKO mice.

How could ER stress then lead to the molecular changes in the OHC taper region? The normal accumulation of ER immediately below the OHC cuticular plate (Mammano et al, 1999) appears to be a favourable location in this regard. We found that *Manf* inactivation distinctly weakened the F-actin cytoskeleton of the OHC cuticular plates, an event that might lead to disturbances in the actin–plasma membrane contacts in the stereocilia taper region. This suggestion is supported by prior data showing that ER stress can evoke actin cytoskeleton remodelling (Van Vliet et al, 2017; Urra et al, 2018).

The expression of PMCA2 and its interacting partner neuroplastin were abolished from the fused stereocilia of OHCs. This suggests that the Ca²⁺ clearance capability was degraded, as OHC stereocilia actively pump Ca²⁺ out of the cell using PMCA2 pumps (Fettiplace & Nam, 2019). Our results do not give evidence that MANF regulates

neuroplastin expression in OHCs, implying that the reported ability of MANF to bind neuroplastin is context-dependent (Yagi et al, 2020). Our results suggest that MANF promotes the integrity of the OHC hair bundle and the compartmentalization of the plasma membrane to distinct functional domains, and when this is abrogated, neuroplastin and PMCA2 expressions and $Ca^{2+}$ clearance are affected.

Despite the prominent defects in the OHC hair bundle, the IHC hair bundle of *Manf* cKO mice lacked obvious morphological alterations. Vice versa, high-frequency IHCs displayed prominent synaptopathy, whereas no ribbon synapse loss was evident in OHCs. These distinct differences in the phenotype caused by *Manf* inactivation might be linked to the differential distribution ER in the cytosol of the two hair cell types (Bullen et al, 2019). Based on CtBP2/Homer1 double-immunostaining, the presynaptic ribbon loss in IHCs was associated with the loss of the corresponding postsynaptic receptor patch, suggesting that the synapse structure as a whole was degraded. If only the presynaptic ribbons had been lost, the remaining synapses might have retained some degree of function; shown to be possible in *RIBEYE* KO mice that have ribbonless IHC synapses (Becker et al, 2018; Jean et al, 2018). The mechanism by which ER stress impairs IHC ribbon synapse maintenance is not understood, but could be due to defects in the production or trafficking of proteins from the ER to the synapse (Martínez et al, 2018; Moser et al, 2020).

Major part of IHC synaptopathy in *Manf* cKO B6 mice took place within a narrow time window, between 3 and 5 wk of age. This is much earlier than the age-related synaptopathy that characterizes the B6 genetic background (Jeng et al, 2020). We found that the remaining CtBP2 puncta in the mutant mice were abnormally large as compared to the non-synaptopathic juvenile mutants and to age-matched controls. Increased volume of presynaptic ribbons has been associated with ageing in B6 mice (Jeng et al, 2020). As ribbon synapse loss in 1-to-2-mo-old *Manf* cKO mice was comparable to aged (15 mo old) control B6 mice (Jeng et al, 2020), the present data point to the importance of the buffering capacity of the ER proteostasis network in antagonizing age-related synaptopathy. Notably, as we did not find synaptopathy in *Manf*-inactivated IHCs of CBA mice, unlike in these cells of B6 mice, it appears that *Manf* deficiency combined with the B6 background carrying the $Cdh23^{ahl}$ mutation is required to build up a sufficient level of ER stress to cause ribbon synapse pathology already at young adulthood. In all, as the synaptopathic end-result shared many similarities to the synaptopathy found in the aged B6 mice, *Manf* deficiency could be described as having accelerated the pathological ageing process of IHCs.

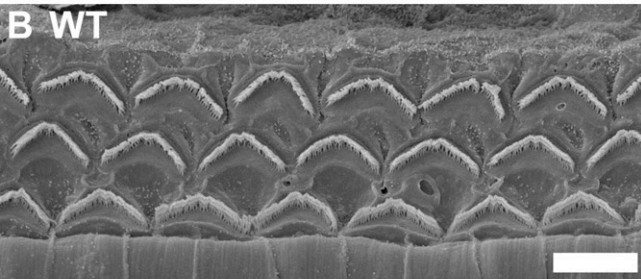

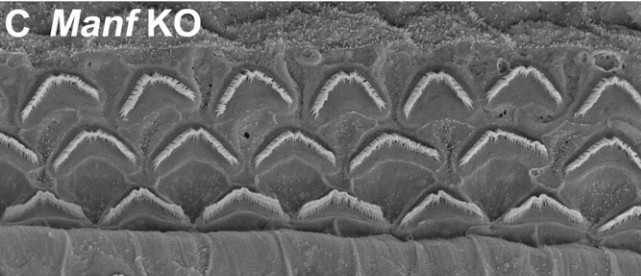

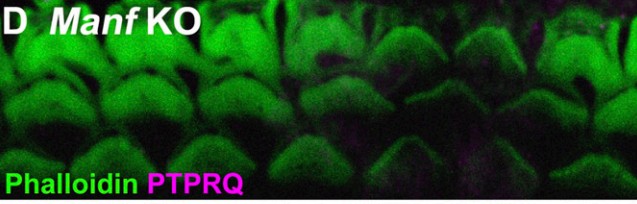

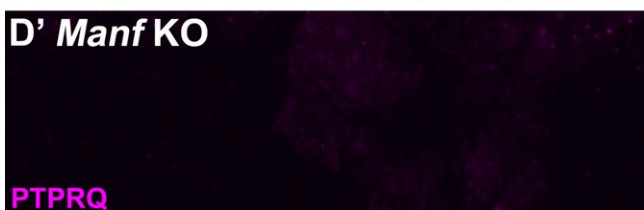

**Figure 7. The strain-dependent genetic background dictates the phenotype of *Manf*-inactivated outer hair cells (OHCs).**
**(A)** ABR thresholds between *Manf* KO and wild type mice under CBA background are without significant differences throughout the frequency region (4–45 kHz), analysed at P56 (compare to age-matched *Manf* cKO B6 mice, Fig 1E). One-way ANOVA P-values: tone stimuli 4 kHz P = 0.458, 8 kHz P = 0.234, 16 kHz P = 0.407, 23 kHz P = 0.700, 32 kHz P = 0.881, 40 kHz P = 0.563, 45 kHz P = 0.632; click stimulus: P = 0.054. Animals per group: *Manf* KO CBA n = 6, wild-type CBA n = 3. Error bars show ± SE. **(B, C)** SEM images from the 45 kHz region of cochleas of WT and KO CBA mice show a comparable OHC hair bundle phenotype. **(D)** PTPRQ immunostaining coupled with phalloidin labeling shows the absence of PTPRQ expression in the 45-kHz OHC hair bundles in *Manf* KO CBA mice, consistent with the normal bundle morphology (compared to age-matched *Manf* cKO B6 mice, Fig 5A–D'). Abbreviations: ABR, auditory brainstem response; B6, C57BL/6J strain; CBA, CBA/Ca strain; cKO, conditional knock out; KO and $Manf^{-/-}$, knock out; OHC, outer hair cell; PTPRQ, protein tyrosine phosphatase receptor type Q; SE, standard error of the mean; SEM, scanning electron microscopy; SPL, sound pressure level; WT, wild type. Scale bar: (B) 5 μm in all images.

**Table 1. The contribution of *Manf* deficiency and *Cdh23^ahl^* mutation to the hearing phenotype in the mouse strains studied.**

| Mouse | *Manf* status | *Cdh23^ahl^* locus sequence | Cdh23 status | Hearing phenotype (P56) |
|---|---|---|---|---|
| C57BL/6J wild type | *Manf +/+* | C C T C C A G T G | *Cdh23^753G→A^* | Normal |
| C57BL/6J Manf cKO | *Manf fl/fl; Pax2-cre* | C C T C C A G T G | *Cdh23^753G→A^* | Severe impairment |
| CBA/Ca wild type | *Manf +/+* | C C T C C G G T G | *Cdh23^753G^* | Normal |
| CBA/Ca Manf KO | *Manf −/−* | C C T C C G G T G | *Cdh23^753G^* | Normal |
| CD-1 wild type | *Manf +/+* | C C T C C A G T G | *Cdh23^753G→A^* | Severe impairment |
| CBAxCD-1 F2 Manf KO | *Manf −/−* | C C T C C A G T G | *Cdh23^753G→A^* | Severe impairment (see also Herranen et al [2020]) |
| CBAxCD-1 F2 Manf KO | *Manf −/−* | Het C C T C C G(A) G T G | *Heterozygous for: Cdh23^753G→A^* | Normal (see also Herranen et al [2020]) |

Each row presents one representative mouse together with its background strain, *Manf* genotype, a snippet of the sequencing chromatogram containing the *Cdh23*^753G→A^ point mutation locus, *Cdh23* genotype, and a basic hearing status at P56. ABRs and outer hair cell loss of CBA x CD-1 hybrid (F2) mice were presented in Herranen et al (2020). *Manf* deficiency and the homozygous *Cdh23^ahl^* mutation are together required for the severe sensorineural hearing loss phenotype (= highly elevated ABR thresholds at all frequencies). Abbreviations: ABR, auditory brainstem response; cKO, conditional knock out; F2, F2 generation; Het, heterozygote; KO, knock out.

B6 mice exhibit early-onset hearing loss, the decline in ABR thresholds starting at 2–3 mo of age (Ohlemiller et al, 2016). This was the reason why we performed analyses in 2-mo-old or younger *Manf* cKO B6 mice. Yet, we were interested to find out if the mutant phenotype, particularly IHC synaptopathy, progressed thereafter. Our unpublished results (K Ikäheimo, U Pirvola) show that the IHC ribbon loss had extended from the basal to the mid-apical part of mutant cochleas by 4 mo of age. However, synaptopathy was also

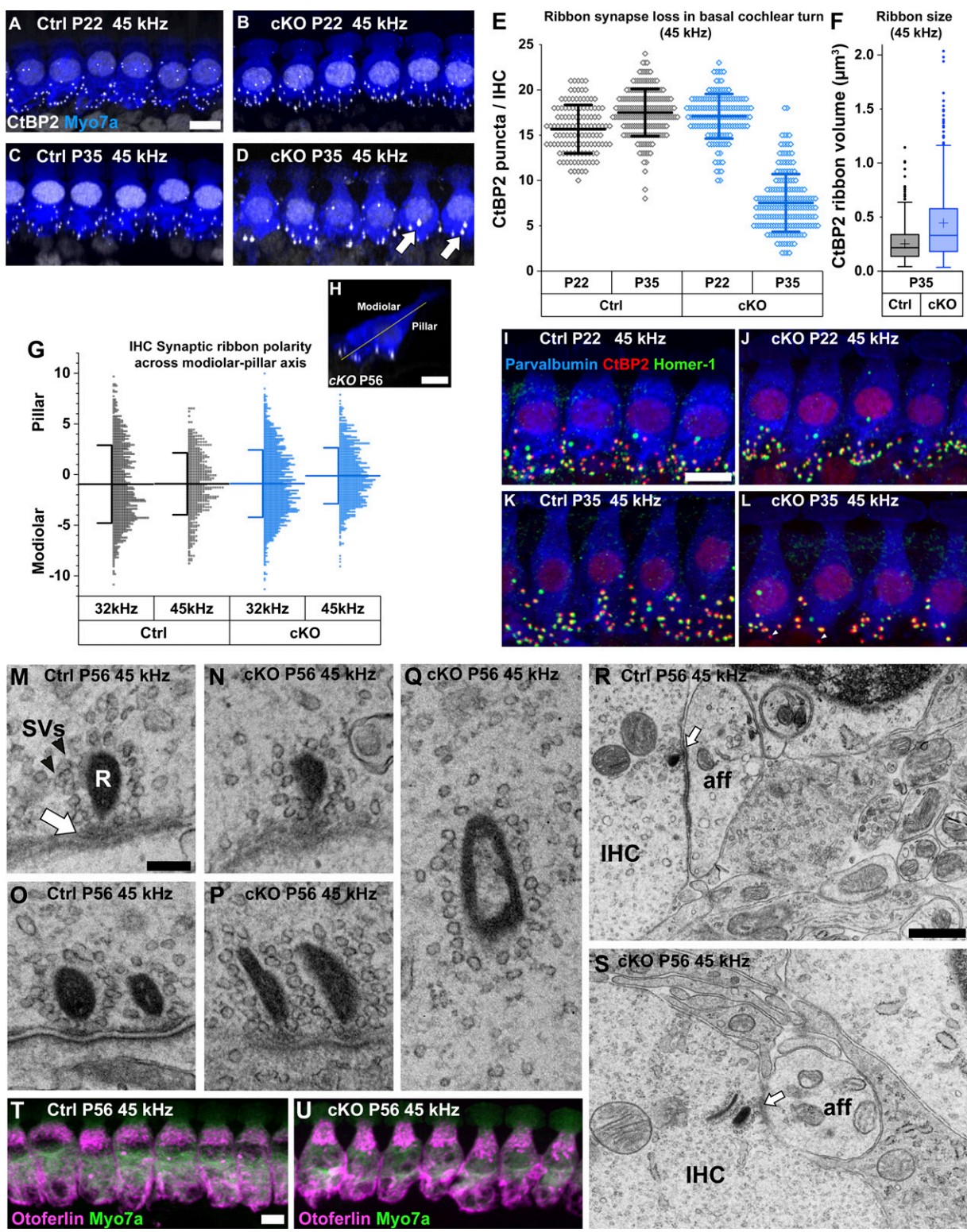

**Figure 8. Pre- and postsynaptic defects in the ribbon synapses of inner hair cells in *Manf*^fl/fl;*Pax2-Cre* B6 mice.**
**(A, B, C, D)** Double-immunostaining for the hair-cell-marker myosin 7a (blue) and the presynaptic ribbon-marker CtBP2 (white puncta) show that the CtBP2 puncta became reduced in number in the 45-kHz inner hair cells (IHCs) of cKO mice between P22 and P35, in contrast to controls. Many of the remaining puncta at P35 appear abnormally large (arrow). IHCs are illustrated as maximum intensity projections of z-stacks spanning from the apical neck region to the basal end of IHCs. **(E)** The two genotypes do not show differences in ribbon numbers in the 45-kHz IHCs at P22, unlike at P35 when about half of ribbons are lost in mutants. P22: *P*-value 0.216 (linear mixed model, type 3 test of fixed effect, genotype); estimated marginal means control 15.581, cKO 16.948; estimate of difference 1.367, control 95% CI [13.713, 17.449], cKO 95% CI [15.331, 18.566]. P35: *P*-value 0.045 × 10⁻³; estimated marginal means control 17.064, cKO 7.618; estimate of difference 9.446, control 95% CI [15.140, 18.989], cKO 95% CI [5.697,

evident in control cochleas. At 4 mo, quantification in the 16 kHz low-frequency region, which was non-synaptopathic in cKOs at 2 mo of age, showed a significant difference in ribbon numbers between the two genotypes. These unpublished results indicate that chronic ER stress drives the progression of age-related synaptopathy along the cochlear base-to-apex (high-to-low frequency) axis.

We found that ER stress-induced IHC synaptopathy lacked a spatial (modiolar versus pillar side of IHCs) preference, unlike synaptopathy following noise exposure (Liberman et al, 2015). Also in this sense the pattern of synapse loss corresponded more closely to what has been observed in ageing mice (Jeng et al, 2020). The modiolar-pillar position of a ribbon synapse is significant, as it corresponds to the neurophysiological properties of innervating spiral ganglion neurons (Taberner & Liberman, 2005; Ohn et al, 2016). In the current study, the actual contribution of the observed synaptopathy to the loss of hearing sensitivity (probed by elevated ABR thresholds) is likely to be minimal because the OHC amplifier was non-functional in the high-frequency region of the cochlea (probed by the lack of DPOAEs). However, if synaptopathic IHCs could still be stimulated to initiate mechanotransduction, for example, by very loud sounds, the synapse loss would likely confer a reduction in the quality of sound information sent to the brain. Thus, the loss of half of the IHC synapses (auditory deafferentation) will diminish the information available to the brain about the acoustic environment (Lopez-Poveda & Barrios, 2013).

In summary, we show that MANF promotes the structural and functional maintenance of auditory hair cells in mice predisposed to genetic age-related hearing loss. Our data are supported by prior studies showing that *Manf* ablation in mice aggravates the pathology of hereditary skeletal dysplasias associated with ER stress (Nundlall et al, 2010; Bell et al, 2019). We show that chronic ER stress is detrimental to the integrity of the domains of hair cells central to hearing function, the OHC stereociliary bundle and the IHC ribbon synapse. We demonstrate early-onset hearing loss in *Manf*-deficient mice and show that it corresponds to the early-onset sensorineural hearing loss found in a patient case with a homozygous loss-of-function *MANF* mutation. Thus far, only two patients carrying *MANF* mutations have been identified and analysed (this study; Yavarna et al, 2015; Montaser et al, 2021).

There is a future need for identification of *MANF* mutations in large clinical databases to understand the progression and severity of this hearing impairment. Altogether, we present *MANF* as a novel gene associated with the hearing function and ER stress as a key pathophysiological mechanism involved in hearing loss.

# Materials and Methods

### Experimental animals

The original *Manf* KO mouse line was maintained under CD-1 background (ENVIGO:030, MGI:5824366, Lindahl et al, 2014). To generate *Manf* KO mice under CBA background, we crossed *Manf* heterozygous CD-1 mice with the CBA/CaOlaHsd strain (ENVIGO:209, MGI:2164153). Repeated backcrossing of hybrid progeny to the CBA strain continued up to F5 and F6 generations. To generate *Manf*<sup>flox/flox</sup>;*Pax2-Cre* cKO mice, we crossed the *Manf*<sup>flox/flox</sup> (Lindahl et al, 2014) and *Pax2-Cre* (Ohyama & Groves, 2004) transgenic mice, both under B6 background (Envigo:043, C57BL/JRccHsd, MGI:6156915). In addition to some other tissues, Pax2 is expressed in the embryonic otic placode that gives rise to the epithelial cell types and neurons of the inner ear (Ohyama & Groves, 2004). Pax2 is not expressed in the pancreatic $\beta$ cells and, thus, these cKO mice are non-diabetic. Control mice included both *Cre*-positive specimens and *Manf* heterozygotes, and wild types. The following primers were used for *Manf*<sup>flox/flox</sup>;*Pax2-Cre* cKO genotyping: 5′-CGT TTT CTG AGC ATA CCT GGA-3′, 5′-AAT CTC CCA CCG TCA GTA CG-3′ (*Cre*), and 5′-TGG AGT GAG CAC AAC TCA GG-3′, 5′-CTC AGG TCC TCC ACA AGA GC-3′ (*Manf*<sup>flox-flox</sup>). The following primers were used for *Manf* KO genotyping: *F*: 5′-TGG AGT GAG CAC AAC TCA GG-3′, *WT*: 5′-GGC TTC GAC ACC TCA TTG AT-3′ and *KO*: 5′-CCA CAA CGG GTT CTT CTG TT-3′.

To find out whether mice used in this study carried the *Cdh23*<sup>753G→A</sup> point mutation (Noben-Trauth et al, 2003), we sequenced the DNA region containing the 753rd nucleotide in the *Cdh23* gene. The following primers were used: Cdh23-F 5′-GATCAAGACAAGACCA-GACCTCTGTC-3′; Cdh23-R 5′-GAGCTACCAGGAACAGCTTGGGCCTG-3′. Sanger

---

9.539]. Animals (cochleas) per group: P22 control *n* = 4, P22 cKO *n* = 5, P35 control *n* = 6, P35 cKO *n* = 5 (20-30 IHCs per cochlea). Data points represent single IHCs. Horizontal bars represents mean of all IHCs in a given group, error bars represent ± SD. **(F)** The volumes of CtBP2-positive presynaptic ribbons from the 45-kHz cochlear region at P35, shown as box-and-whiskers plots. Box covers the second quartile, whiskers extend to 1.5× the interquartile range. Horizontal line denotes the median and cross denotes the mean. Outliers are shown as individual data points. The distribution is significantly different between the genotypes (Kolmogorov–Smirnov test, *P* < 0.001, *n* = 3 animals per group, 823 and 397 synapses analysed in the control and cKO groups, respectively). **(G, H)** The positions of ribbon synapses across the IHC modiolar-pillar axis, illustrated in G, were quantified at P56. In the synaptopathic (45 kHz) region of mutant cochleas, there was a small shift in mean ribbon position toward the pillar side, but variability within the groups was high and the difference between the genotypes did not reach significance. A non-synaptopathic region (32 kHz) did not show any shift or difference in mean ribbon positions between control and mutant mice. 32 kHz: *P*-value 0.582 (linear mixed model, type 3 test of fixed effect, genotype); estimated marginal means control −1.051, cKO −0.809; estimate of difference 0.242, control 95% CI [-1.565, −0.537], cKO 95% CI [-1,180, −0.437]. 45 kHz: *P*-value 0.263; estimated marginal means control −0.963, cKO −0.145; estimate of difference 0.816, control 95% CI [-2,247, 0.321], cKO 95% CI [-1,261, 0.972]. Animals (cochleas) per group: Control *n* = 3, cKO *n* = 4. Horizontal bars represent the mean value of pooled measurements from all mice in a given group, error bars represent ± SD. Data points represent the distance of single ribbons from the modiolar-pillar axis (in μm). Negative sign denotes modiolar side, positive denotes pillar side. **(I, J, K, L)** Triple-immunostaining for parvalbumin (hair cell marker, blue), CtBP2 (red), and Homer1 (post-synaptic receptor marker, green) shows that the 45-kHz IHCs in mutant mice that lose ribbons also lose at least an equal amount post-synaptic receptors between P22 and P35 (K). Unpaired "orphan" ribbons (arrowhead) were often seen in the IHCs of P35 mutant mice. Quantification of these findings is detailed in results. **(M, N, O, P)** TEM sections show that the high-frequency IHCs of both genotypes exhibit single as well as double ribbons at the synaptic active zone (arrow), with numerous synaptic vesicles (arrowhead) tethered to the ribbon plaque. **(Q)** Ribbons surrounded by synaptic vesicles can be found "free-floating" in the IHC cytoplasm in P56 cKO mice. **(R, S)** In both controls and mutants at P56, the terminals of spiral ganglion neurons at the basal end of high-frequency IHCs lacked signs of swelling. **(T, U)** Double-immunostaining for myosin 7a (green) and otoferlin (magenta) shows comparable otoferlin expression in the 45-kHz IHCs in the two genotypes at P56. IHCs are illustrated as maximum intensity projections of *z*-stacks spanning the entire IHCs. Abbreviations: aff, afferent nerve terminal; B6, C57BL/6J strain; CI, confidence interval; cKO, conditional knock out; CtBP2, C-terminal Binding Protein 2; Ctrl, control; IHC, inner hair cell; R, ribbon plaque; SVs, synaptic vesicles; TEM, transmission electron microscopy. Scale bars: (A) 5 μm A-D; (G) 5 μm G; (H) 5 μm H-K; (L) 100 nm L-P; (Q) 500 nm Q, R; (S) 5 μm S, T.

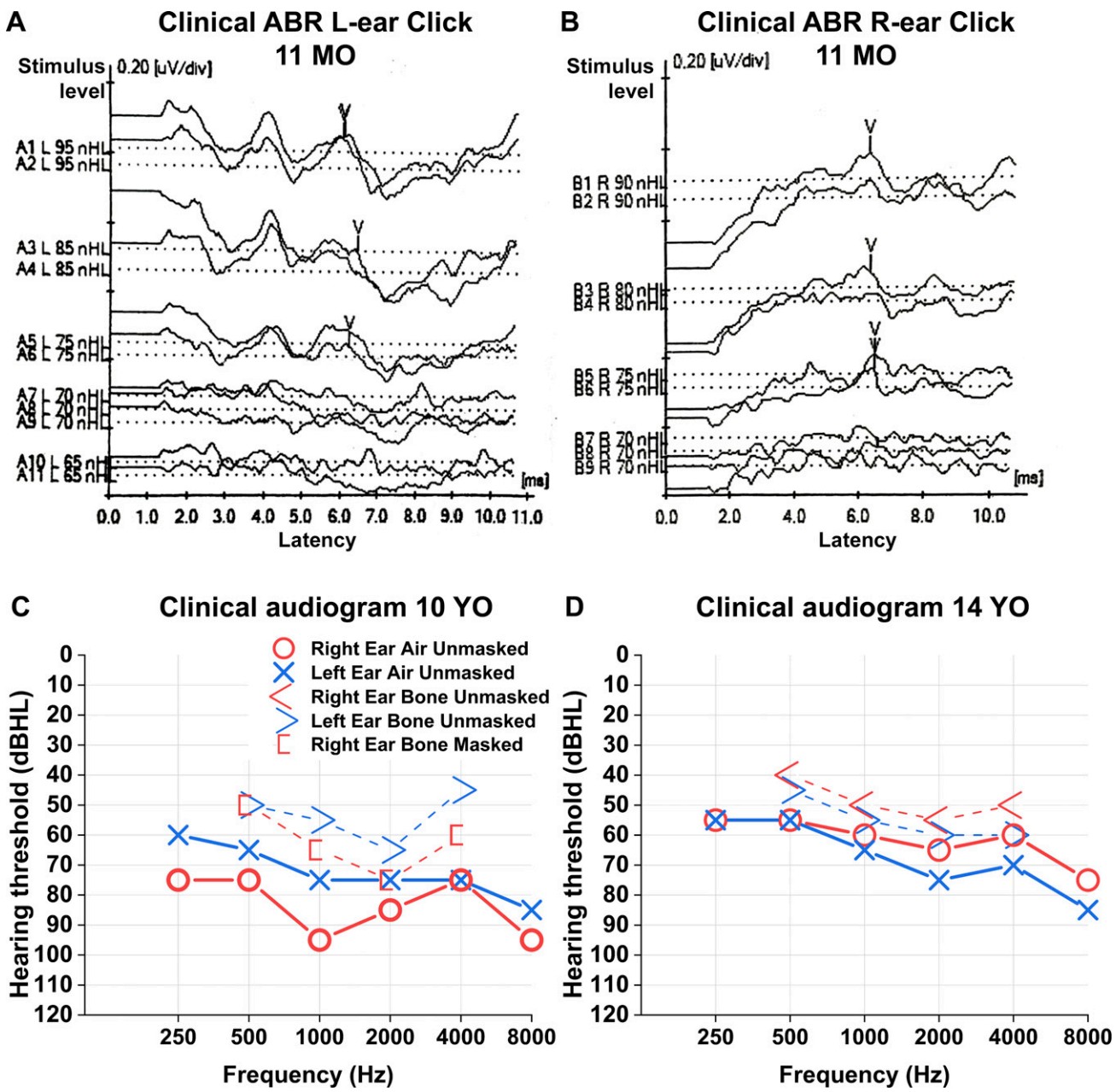

**Figure 9. Hearing impairment in a patient with homozygous loss-of-function variant of *MANF*.**
**(A, B)** Patient's click ABR results at the age of 11 mo. Wave complex IV-V can be seen with 75 dBHL stimulus level (y-axis) on both sides. However, at 70-dBHL stimulus level, the wave complex is unclear. **(C, D)** Clinical audiograms at the age of 10 yr (C) and 14 yr (D) show severe symmetric bilateral sensorineural hearing loss. See text for details. Abbreviations: ABR, auditory brainstem response; dBHL, decibels hearing level; MO, months old; nHL, normalized hearing level; YO, years old.

sequencing was performed at the DNA Sequencing and Genomics core facility at the University of Helsinki. Staden package's Gap4 was used for sequence analysis. Both females and males were included in the groups of mice analysed.

**ABRs**

ABRs were acquired as described previously (Herranen et al, 2020). All equipment was from Tucker-Davis Technologies (TDT System 3).

Thresholds were determined by visual inspection for the lowest sound intensity at which a consistent, repeatable waveform could be obtained. Calibration was performed with PCB-378C01 calibration microphone with a model 480C02 sensor signal conditioner (PCB Piezotronics Inc.) using BioSigRZ and RPvdsEx virtual design studio (version 84, TDT) software. We anaesthetized mice via intraperitoneal injections of ketamine hydrochloride (75 mg/kg, Ketaminol, Intervet International Inc.) and medetomidine hydrochloride (1 mg/kg, Domitor, Orion Corporation).

## Distortion product otoacoustic emissions

DPOAE recordings were performed immediately following ABR recordings. A booster injection of the anaesthetic (halved dose) was administered and the mice were kept on a warming water bath during the measurements. Inferior part of the pinna, occluding direct line of sight into the ear canal, was minimally cut to allow the DPOAE recording probe a straight approach. TDT system 3 with the RZ6 Multi I/O processor was used for stimuli generation (SigGenRZ software), stimuli presentation, and for the digital sampling and analysis of responses (BioSigRZ software). The recording microphone (ER-10B+, Etymotic Research Inc.) was placed close to the ear canal opening, with an additional 3 mm ear tip (ER10D-T03) enclosing the space between the canal and the probe. Two MF1 Multi-field Magnetic Speakers (speaker 1 and 2) were used to deliver probing stimuli at 8–32 kHz (geometric mean), where the frequency ratio between the stimuli was 1.2 = $F_2/F_1$ and the intensity difference was 10 = $L_1 − L_2$ (in dB sound pressure level, SPL) throughout the experiment. Calibration was performed in-ear for each probe insertion. Stimuli were presented from 10 to 70 dB SPL ($L_2$) in 5 dB SPL steps. 128 responses were averaged. From fast Fourier transforms, the $2F_1 - F_2$ frequency place (DPOAE) amplitude was analysed, with the noise floor extracted from a 1,000 Hz wide frequency band centred at $2F_1 - F_2$. The DPOAE amplitude had to exceed the noise floor by two SD to be interpreted as a valid DPOAE. For each probed frequency, the lowest $L_2$ that elicited a valid DPOAE was interpreted as the DPOAE threshold level.

## Light microscopy sample preparation

We perilymphatically perfused cochleas by gentle flushing with 4% PFA, pH 7.4, in PBS, followed by immersion in this fixative for 2 h at +4°C. Cochleas were decalcified with 0.5 M EDTA, pH 7.5, overnight at +4°C. Cochleas were dissected to produce whole-mount preparations with the organ of Corti exposed. Whole mounts were blocked with 10% goat serum (Jackson ImmunoResearch) in PBS containing 0.25% Triton-X-100 (PBS-T) for 1 h at RT. Specimens were incubated with the primary antibody cocktail in PBS-T for 48 h at +4°C, and with the secondary antibody cocktail similarly for 24 h at +4°C.

We used the following primary antibodies in double- or triple-labeling: Rabbit polyclonal radixin (Cat. no R3653, RRID: AB_261933, used at 1:250; Sigma-Aldrich), Rabbit polyclonal Homer1 (Cat. no. 160003; RRID:AB_2832230, used at 1:2,000; Synaptic Systems), sheep polyclonal neuroplastin (Cat. no. AF7818, RRID:AB_2715517, used at 1:500; R&D Systems), rabbit polyclonal PMCA2 (Cat. no. ab3529, RRID:AB_303878; Abcam), goat polyclonal parvalbumin (Cat. no. PVG-213, RRID:AB_2650496, used at 1:1,500; Swant), rabbit polyclonal myosin 6 (Cat. no. 25-6791, RRID: AB_10013626, used at 1:1,000; Proteus Biosciences), rabbit polyclonal myosin 7a (Cat. no. 25-6790, RRID: AB_10015251, used at 1:3,000; Proteus Biosciences), mouse monoclonal C-terminal Binding Protein 2 (CtBP2; BD Biosciences Cat. no. 612044, RRID: AB_399431, 1:200), mouse monoclonal otoferlin (Cat. no. ab53233, RRID: AB_881807, used at 1:500; Abcam). Affinity purified rabbit polyclonal PTPRQ antibody was a kind gift from Guy Richardson, University of Sussex. Primary antibodies were detected using Alexa Fluor 488/594/647-conjugated goat anti-rabbit/mouse IgG secondary antibodies (Invitrogen). After antibody incubations, we visualized F-actin filaments using Oregon Green 514- or Rhodamine 568-conjugated phalloidin (Invitrogen). Nuclei were stained with DAPI. ProLong Gold anti-fade reagent was used for mounting (Invitrogen).

## Light microscopy imaging

All light microscopy images were obtained with the Axio Imager.M2 microscope equipped with Apotome 2 structured illumination microscopy (SIM) slider, using PlanApo 10× (NA = 0.45), PlanApo oil-immersion 40× (NA = 1.30), and PlanApo oil-immersion 63× (NA = 1.40) objectives (all from Zeiss). The Hamamatsu ORCA Flash 4.0 V2 camera and ZEN 2 software (Zeiss) were used for image acquisition.

All immunofluorescence images are representative takes of respective experimental groups (three mice per group, one cochlea per mouse) in two or more independent experiments, and all immunofluorescence quantifications were performed with at least three different samples per group.

## Electron microscopy sample preparation

Cochleas were perilymphatically fixed with 2.5% glutaraldehyde in 0.1 M phosphate buffer (pH 7.2) and then immersed in this fixative for 48 h at +4°C. For SEM imaging, intact tectorial membranes were folded off to the side of the organ of Corti during dissection. SEM specimens were post-fixed with 1% osmium tetroxide for 1 h at RT. They were then were dehydrated with a graded series of ethanol (70%, 96%, and 100%). Juvenile and adult (P15 and older) SEM specimens were dried using hexamethyldisilatzane, then mounted and coated with a thin layer of platinum (Quorum Q150TS coater). To optimally preserve inter-stereociliary links, P9 were not treated with osmium tetroxide, but were dehydrated by a graded series of ethanol (as above), followed by critical point drying in $CO_2$ (Hadi et al, 2020). TEM specimens were treated with osmium tetroxide as above and incubated with transitional solvent acetone and embedded in epoxy resin (Epon, TAAB Laboratories Equipment Ltd).

For TEM analysis of the hair cell stereocilia bundle and cuticular plate, resin-embedded specimens were sectioned in transverse orientation to 60 nm thick sections in 1–5-µm intervals. For TEM analysis of ribbon synapses, sections of the same thickness were cut in horizontal orientation in 2-µm intervals to catch more synapses in each section. For pre- and postsynaptic pairing analysis, we produced multiple consecutive thin sections per sampling interval, so that care was taken to determine whether a presynaptic ribbon was apposed to a postsynaptic density. TEM sections were post-stained with uranyl acetate and lead citrate.

## Electron microscopy imaging

### SEM

FEI Quanta 250 Field Emission Gun Scanning Electron Microscope (FEI) with Everhardt Thornley secondary electron detector was used to acquire SEM images of cochlear whole mount specimens. Images were captured under high vacuum with a 3–5 kV beam and with a spot size of 2.6–3.0 depending on magnification and sample orientation. For the inter-stereociliary link analysis in P9 whole mount specimens, ZEISS Crossbeam 550 (Zeiss) was used with an accelerating voltage of 1.2 kV and probe current ranging from 80 to 300 pA.

### TEM

Jeol JEM-1400 (Jeol) microscope was used to analyse the thin sections using an 80 kV beam. Gatan Orius SC1000B bottom mounted CCD-camera (Gatan Inc.) was used for image acquisition.

All electron microscopical images are representative takes of respective experimental groups (three mice per group, one cochlea per mouse), except SEM imaging of P35 control OHC hair bundles where only two mice were studied.

All electron microscopy studies were performed at the Electron Microscopy Unit (EMBI) core facility at the University of Helsinki.

### OHC cuticular plate phalloidin intensity quantification

We quantified the intensity of rhodamine phalloidin (probes for F-actin) in the OHC cuticular plates. For each experiment, mutant and control cochleas were fixed, decalcified, dissected and stained in parallel. Likewise, image acquisition was performed using identical hardware and software settings. z-stacks were obtained with Axio Imager.M2 microscope equipped with Apotome 2, using PlanApo oil-immersion 63× (NA = 1.40) objective.

To use structured illumination imaging data for quantitative fluorescence intensity analysis, we made use of the QSIM algorithm, a plugin for ImageJ (Gao, 2015). Raw data from Zeiss Apotome SIM (three phase images), was processed by QSIM, resulting in voxel values in photons instead of arbitrary grey values (Hagen et al, 2012).

The cuticular plate was identified manually from each OHC by their staining pattern. Only the OHCs imaged upright (stereocilia pointing toward the objective) were qualified for analysis. Any above-background signal above the cuticular plate also disqualified the OHC from analysis for the purpose of minimising contribution of noise to the analysis of the SIM images. A 2 $\mu$m diameter circular area in the middle of each cuticular plate was outlined for the purpose of avoiding signal spread from the stereocilia or adjacent supporting cells. Mean photon count was calculated from this area in five consecutive images (step size 0.25 $\mu$m) to get a comparative measure of the mean intensity originating from the ~1 $\mu$m thick cuticular plate. For data presentation, we scaled cuticular plate intensities by dividing by the total mean OHC cuticular plate intensity from the control preparation(s) of each separate experiment to account for any experiment-to-experiment variability in the overall staining intensity.

### Presynaptic ribbon synapse counts and modiolar-pillar position

To locate particular frequency regions in the cochlea, we prepared a frequency map from each whole mount using the MeasureLine.class ImageJ plugin (Eaton-Peabody Laboratories Histology Core resources, https://www.masseyeandear.org/research/otolaryngology/eaton-peabody-laboratories/histology-core) and calculations according to Müller et al (2005). The CtBP2 antibody was used to label the synaptic ribbons and either myosin 6 or myosin 7a antibodies were used to label IHC bodies and to segment and group ribbons between IHCs. Synaptic ribbon quantification per IHC was performed from 16, 32, and 45 kHz regions by manually annotating positions (x, y, and z) of the CtBP2 puncta image-by-image in the image stacks in

ImageJ. ~30–40 IHCs were captured in each image stack from a frequency region in each cochlea. z-stacks were obtained with Axio Imager.M2 microscope equipped with Apotome 2, using PlanApo oil-immersion 40× objective (NA = 1.30). Each stack spanned from the cuticular plate to the synaptic pole of IHCs, with a z-step size of 0.25 $\mu$m, capturing the entire IHC body.

To measure CtBP2-positive ribbon synapse volumes, we used the manually annotated synaptic puncta positions, 3D black-on-white thresholding, and a 3D object volume separation algorithm (4/6 pixel connectivity) to segment and measure the ribbon volumes in image stacks (Microscopy Image Browser v.2.81, Belevich et al, 2016). To mitigate comparability issues due to arbitrary thresholding, we standardized the thresholding step between samples by using the nucleic staining of CtBP2 in the organ of Corti as a reference under the assumption that the nucleic CtBP2 expression remained comparable between experimental and control groups.

To identify and measure ribbon synapse polarity and distance from the IHC modiolar-pillar axis, we used the annotated positions of the CtBP2 puncta in image stacks to compute Euclidean distances to a plane dividing each IHC body to "modiolar" (closer to the modiolus) and "pillar" (closer to supporting pillar cells) halves. FIJI ImageJ toolkit was used to define the axis for each IHC and visually validate the definition of the modiolar and pillar sides, and MATLAB was used to compute distances, in a custom algorithm. First, in FIJI, all voxel dimensions were made to match (isometric stack) using ResliceZ. All relevant image data (IHC cell body label and ribbon coordinates, in separate channels) were interpreted to the new 3 d space by the same algorithm. Following, we manually defined a transverse reslicing for each IHC by their longitudinal axis using the reslice command. The contours of the IHC side-profiles were manually outlined based on the cell body label's grey values being above an arbitrary threshold, drawing an area (IHC projection from the side) with visible modiolar-pillar sides. To define the axis in a consistent way, the IHC side projection area was selected and we used the best-fitting ellipse routine of the Fit Ellipse command. The major axis of the ellipse was defined as the modiolar-pillar axis. The major axis end point coordinates and ribbon coordinates were extracted and transferred to MATLAB. The end point coordinates were used to describe the modiolar-pillar plane in 3 d space for each IHC (the resliced stack with corresponding ribbon positions) using MatGeom library's createPlane function by Dr. David Legland (https://github.com/mattools/matGeom) (Legland, 2016). Finally, the ribbon coordinates were used together with the defined plane to calculate the shortest distance and its sign between each ribbon and the plane using MatGeom's distancePointPlane function. The sign of an output value defined a ribbons modiolar-pillar assignment, where negative sign stood for the modiolar side. Resulting table of values was exported to SPSS and Origin Pro for further analysis and data visualization (see the Statistical analysis section).

### Statistical analysis

SPSS Statistics (v.25.0.0.1; IBM) was used for all statistical analyses. All data graphing was performed using OriginPro 2020 (v. 9.7.0.188; OriginLab Corporation). Error bars represent SD except in ABR and DPOAE measurements where standard error of mean (SE) is used. Binomial test was used to test whether the proportion of CtBP2

puncta missing the paired Homer1 puncta in the mutant mice group differs from the proportion found in the age-matched control mice group. One-way ANOVA was used for means comparisons between experimental groups and control animals in ABR and DPOAE thresholds. Kolmogorov-Smirnov test was used to compare the distributions of CtBP2-positive ribbon synapse volume measurements between genotypes. For cuticular plate F-actin intensity quantification, ribbon synapse (CtBP2) and post-synaptic receptor (Homer1) per IHC quantification, and ribbon synapse position across IHC modiolar-pillar axis quantification, a linear mixed model (MIXED procedure, SPSS) was used. For phalloidin intensity analysis in the cuticular plate, we included the contribution of (1) genotype and (2) immunostainings performed on separate days (fixed effects), as well as (3) variation between individual mice (random effect) in the model. To fulfil the assumption of normal distribution of data for the model, we analysed the $\log_2$ intensities of the phalloidin signal. For ribbon synapse and post-synaptic receptor per IHC analysis, we included the contribution of (1) genotype and (2) mouse age (fixed effects), as well as (3) variation between mice (random effect) in the model. For ribbon synapse modiolar-pillar position analysis, we included the contribution of (1) genotype (fixed effect), and (2) variation between mice as well as (3) variation between IHCs (random effects) in the model. We report $P$-values, which represent the significance of genotype (type III test of fixed effect), together with the estimated marginal means, the estimate of the difference between genotypes, and the 95% confidence intervals (CI).

## Supplementary Information

## Acknowledgements

This work was funded by The Finnish Ministry of Defence (MATINE) (U Pirvola), Genelec Inc. (U Pirvola), Finnish Association of Otorhinolaryngology and Head and Neck Surgery (U Pirvola), Finnish Hearing Association (U Pirvola), Jenni and Antti Wihuri Foundation (K Ikäheimo), Jane and Aatos Erkko Foundation (M Saarma), Academy of Finland (Grant no. 310891, M Saarma; No 333974, M Lindahl; No. 117044, M Saarma and M Lindahl) and Wellcome Trust post-doctoral fellowship (110082/Z/15/Z, KA Patel) and Career Development Fellowship (219606/Z/19/Z, KA Patel). We thank Guy Richardson (University of Sussex) for kindly gifting PTPRQ antibody for this work. We thank Ilya Belevich (University of Helsinki) for the invaluable help in EM imaging. We acknowledge Sanna Sihvo for the excellent technical assistance provided and Tommi Anttonen for providing feedback on the manuscript. We acknowledge NIH Knock-out Mouse Program (KOMP) for the *MANF*-targeted ES cell clone used to generate the *MANF* KO mice.

### Author Contributions

K Ikäheimo: conceptualization, data curation, software, formal analysis, validation, investigation, visualization, methodology, and writing—original draft, review, and editing.
A Herranen: conceptualization, formal analysis, investigation, methodology, and writing—review and editing.
V Iivanainen: investigation and methodology.
T Lankinen: investigation and writing—review and editing.
AA Aarnisalo: data curation, investigation, and writing—original draft.
V Sivonen: methodology.
KA Patel: resources, data curation, and writing—original draft.
K Demir: resources and investigation.
M Saarma: resources and writing—original draft.
M Lindahl: resources and writing—original draft.
U Pirvola: conceptualization, resources, data curation, software, formal analysis, supervision, funding acquisition, validation, investigation, visualization, methodology, project administration, and writing—original draft, review, and editing.

### Conflict of Interest Statement

M Saarma is an inventor in a MANF-related patent owned by Herantis Pharma Plc.

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
