## [Reviewer comments · Life Science Alliance]

Life Science Alliance

MANF supports the inner hair cell synapse and the outer hair cell stereocilia bundle in the cochlea

Kuu Ikäheimo, Anni Herranen, Vilma Iivanainen, Tuuli Lankinen, Antti Aarnisalo, Ville Sivonen, Kashyap Patel, Korcan Demir, Mart Saarma, Maria Lindahl, and Ulla Pirvola

DOI: <https://doi.org/10.26508/lsa.202101068>

Corresponding author(s): Ulla Pirvola, University of Helsinki and Kuu Ikäheimo, University of Helsinki

Review Timeline:

Submission Date:	2021-03-16
Editorial Decision:	2021-05-20
Appeal Received:	2021-05-26
Editorial Decision:	2021-06-01
Revision Received:	2021-10-08
Editorial Decision:	2021-11-01
Revision Received:	2021-11-08
Accepted:	2021-11-08

Transaction Report:

May 20, 2021

Re: Life Science Alliance manuscript #LSA-2021-01068-T

Dr. Ulla Pirvola
University of Helsinki
Institute of Biotechnology
P.O. Box 56
Helsinki 14
Finland

Dear Dr. Pirvola,

Thank you for submitting your manuscript entitled "MANF, an ER homeostasis regulator, promotes stereocilia and synapse integrity in auditory hair cells" to Life Science Alliance. The manuscript has now been seen by expert reviewers, whose reports are appended below.

Please accept our sincerest apologies for this extended and unusual delay in getting back to you. As you will see from the reviewers' comments below, the reviewer opinions were mixed, and thus, led to a detailed editorial discussion on our end. Unfortunately, after an assessment of the reviewer feedback, our editorial decision is against publication in Life Science Alliance.

While all the 3 reviewers seemed impressed by the technical strength of the manuscript, they also raise a number of fundamental issues that preclude us from considering the study further, #1, The advance of this manuscript over your previous publication, Herranen et al, 2020, remains very limited. While confirmatory studies are acceptable at LSA, we agree with the reviewer that the two publications are very similar with a lot of overlap in data and takehome message, #2, Reviewer 2 points out that the Manf-KO phenotypes could be secondary to a defect in embryonic development in general, and that Manf-KO mice have not been bred for enough numbers of generations (at least 10 would be required) to make it a pure-bred, #3, Since the paper is a mechanistic follow-up of the previous study, the point made by Reviewer 3 that all of this detail does not necessarily provide meaningful physiological insight into the mechanism by which Manf essentially is potentiating the already-known effect of the Cdh23 Ahl allele is also of concern.

We are sorry our decision is not more positive, but hope that you find the reviews constructive. Of course, this decision does not imply any lack of interest in your work and we look forward to future submissions from your lab.

Thank you for your interest in Life Science Alliance.

Sincerely,

Shachi Bhatt, Ph.D.
Executive Editor
Life Science Alliance
<http://www.lsajournal.org>
Tweet @SciBhatt @LSAjournal

Reviewer #1 (Comments to the Authors (Required)):

1. Multiple mechanisms are associated with normal function of the inner ear, and pathogenesis leading to deafness. One of these, which is understudied, is the role of the endoplasmic reticulum (ER) in hearing and ER stress associated with deafness. Given the sensitivity of hair bundles and ribbon synapses to stress, and the ubiquity of ER stress in many diseases' pathogenesis, this is a particular compelling area to study. The work provides an advance in this area, outlining in detail the effects of ER stress and contribution to hair cell damage and the pathophysiology of deafness. The authors studied the effects of MANF deficiency in mice and the associated connection to ER stress. They showed that ER proteostasis impairment triggers early-onset hair bundle changes, which is responsible for the hearing abnormalities. Interestingly, differences in genetic background have proven to be a key element in variability of hearing levels in mice. This is mostly due to the Cdh23 allele in C57Bl/6 mice, which leads to hearing loss. It also exists in CD-1 mice, and the authors' results suggest that ER stress contributes to the hair cell pathology and hearing loss. They further found that ER stress leads to synaptopathy, associated with the loss of inner hair cells. The authors also added the audiological characterization of a patient with a recessive form of MANF-associated deafness.

2. The work done is detailed and extensive, with support of the hypotheses presented.

3. Additional issues to be addressed

Line 42

We propose the involvement of this mechanism in the pathophysiology of Usher syndromes.

There is no evidence that suggests this mechanism is involved in US, which is a combination of deafness and RP. The only phenotype described in the manuscript is deafness. Please remove and replace "Usher syndromes" with "deafness." While *Cdh23* is associated with Usher syndrome, there is no additional evidence regarding this syndrome, rather than deafness alone.

Line 189

Remove 'data not shown',

Either show the data in supplementary or take out this sentence. It is no longer acceptable to make this statement.

I am unable to access Montaser et al paper without payment; therefore, it is hard to judge if the data on the human patient is novel, or had already been described in the previous manuscript.

Reviewer #2 (Comments to the Authors (Required)):

The study by Ikäheimo et al investigates how endoplasmic reticulum (ER) stress elicits hair cell pathology, using mouse models with inactivation of *Manf* (Mesencephalic astrocyte-derived neurotrophic factor). MANF is an endoplasmic reticulum protein that promotes ER homeostasis and has a cytoprotective function. When *Manf* was conditionally knocked out in *Pax2* cre mice this caused hearing loss from early stages that was likely to result from abnormal OHCs. The OHCs had disorganised stereocilia, being worse affected in the basal coil. The hearing loss progressed as did the OHC stereocilia deformity, showing fusion of stereocilia and OHC death. By contrast the IHCs had normal bundles but lost up to 50% of ribbon synapses in the basal coil. All of these abnormalities were correlated with genetic background, only being present in mice with a *cdh23* mutation. This was assumed to be due to the extra ER stress in these animals that is pushed over the limit by *manf* dysfunction.

Finally they show a human with similar deafness due to mutation in *manf*.

The ms is well written and the experiments are performed well. I just have a few main questions.

1. The previous publication from the lab is very similar to this study with a lot of overlap. I understand that the mice have been investigated at an earlier stage but I think the main messages of the papers remain very similar. The study is fairly descriptive and does not offer any mechanistic advance on what is already established.
2. What cells does the *pax2* cre target? There is no explanation or characterisation of *manf* expression or lack of it.
3. Are the effects of *manf* KO really related to a direct effect of *manf* or is it a developmental indirect effect? For instance, could it be something developmental related to mutation in *cdh23*? This needs to at least be clearly explained in the ms i.e. that all the effects could be secondary to some developmental defect.
4. The mouse genetic background of the KO seems quite a mess. Five or six generations are not enough to get pure breed, at least 10 generations are required. This should be explicitly explained.
5. The controls should also be cre positive with non-mutant *manf*. This would ensure that the *Pax2* cre expression is not having any direct effect.
6. Have the afferents and ribbon synapses been characterised in the OHCs? This would be a good comparison to make with the IHCs. Are the mechanisms affecting the synapses in both hair cell types similar or not?
7. The information on the human study is very limited. There is nothing mentioned of any other underlying defect or mutation that could bring the *manf* mutation into effect. The author's show that the *manf* KO has no affect by itself so what is going on in this patient?
8. The IHC controls in figure 3 should be shown since the expression patterns seem to be different to the OHCs. This should also be explained.
9. In figure 5N why is only one mouse used? Why not use both mice for this? The numbers of mice are very low anyway at n=2 for the analysis of TEM ribbon morphology so at least use all you have. Were the ribbons in CD-1 mice investigated at all? It would be interesting to see if the numbers are comparably low, although I understand there is no control for these.
10. If you are drawing conclusions from data then they should be shown - line 173-179

11. The figures should be labeled in a way that would make them easier to read. Write what the colors of the staining represent, label what is in the picture i.e. WT or cKO etc. Also label the axes on the human hearing data.

Reviewer #3 (Comments to the Authors (Required)):

In this study, the authors expand on their previous finding that mice deficient in *Manf*, an ER stress mediator, develop hearing loss. They show evidence of stereociliary disruption and, ultimately, fusion in outer hair cells as well as loss of ribbon synapses in inner hair cells. They demonstrate that this phenotype is likely dependent on the presence of *Cdh23* Ahl allele, which is strongly associated with rapid age-related hearing loss due to disruption of stereociliary tip links. Finally, they provide evidence from a single human subject that *Manf* deficiency is related to hearing loss in humans. This study provides important corroboration for the role of *Manf* (and, by extension, ER stress) in hearing loss.

However, one critical timing problem limits the impact of their conclusion. Specifically, they show that *Manf* cKO mice have significant hearing loss, particularly at high frequencies, as early as P15 (shortly after the onset of hearing). In fact, the high-frequency thresholds are largely the same at P15 and P56. However, they find evidence of subtle stereociliary splaying in both high- and low-frequency outer hair cells at P22 (though they do not show the images for the low-frequency OHCs), and only evidence of stereociliary fusion at P56. Therefore, the time course and frequency specificity of their stereociliary phenotype (splaying and/or fusion) does not match with the timing of the hearing phenotype. This calls into question any conclusion they may draw into the specific mechanistic role through which *Manf* deficiency causes hearing loss through structural changes in the stereocilia, which is a central point of the paper.

The other points of the paper - the apparent synergy of the Ahl allele and *Manf*-induced hearing loss; and the case report of the human with *Manf* variant and hearing loss - are interesting. Though the Ahl/*Manf* finding should be discussed in contrast with the previous finding from Li et al (JCI, 2018) that *TMTC4*, another ER-stress-associated hearing-loss gene, still had a rapid progressive hearing loss phenotype even when outcrossed from B6 to FVB, suggesting that the *Cdh23* Ahl allele is not required for ER-stress-associated hearing loss.

Overall, the authors provide a very technically strong, detailed characterization of the stereociliary changes that occur in their *Manf* cKO. However, all of this detail does not necessarily provide meaningful physiological insight into the mechanism by which *Manf* essentially is potentiating the already-known effect of the *Cdh23* Ahl allele, and is therefore of somewhat limited impact.

*Re: Life Science Alliance Manuscript - Editorial Decision
LSA-2021-01068-T

Dear Editor-in-Chief,

Thank you very much for getting back to us. We are of course disappointed by your decision not to allow us to prepare a revised ms. The comments by reviewers were constructive and we can address their criticism. We do not agree with the comment of extensive similarity with our previous paper. Therefore, we would be grateful if you reconsider your decision, based on our comments below.

Editor comment #1: The advance of this manuscript over your previous publication, Herranen et al, 2020, remains very limited. While confirmatory studies are acceptable at LSA, we agree with the reviewer that the two publications are very similar with a lot of overlap in data and takehome message.

Author response: The criticism of similarity between the ms and our published paper was raised by *one out of three reviewers*. In the published paper, we showed in adult mice that *Manf* inactivation leads to hearing loss and that it is associated with hair cell death. While it is true that hair cell death is a part of the mutant phenotype, the current ms shows that *non-functionality of hair cells* is the primary reason for the severe hearing deficit and that the non-functionality is present already at the age when hearing function commences. We show *novel molecular data of how ER stress perturbs the integrity of the mechanotransduction domain (hair bundle) and of the hair cell synapse (ribbon synapse),* these being separate events from the activation of proapoptotic pathways. Of course, because the same mutant mouse model was analyzed, the final manifestation of the mutant phenotype-hearing loss and hair cell death-remains the same in the ms and the published paper. In all, *the ms provides new insight of the molecular events and genetic status that are associated to the hearing loss phenotype.*

Editor comment #2: Reviewer 2 points out that the Manf-KO phenotypes could be secondary to a defect in embryonic development in general, and that Manf-KO mice have not been bred for enough numbers of generations (at least 10 would be required) to make it a pure-bred.

Editor comment #3: Since the paper is a mechanistic follow-up of the previous study, the point made by Reviewer 3 that all of this detail does not necessarily provide meaningful physiological insight into the mechanism by which Manf essentially is potentiating the already-known effect of the Cdh23 Ah1 allele is also of concern.

Author response: Reviewer 2 did not suggest a possible contribution of *Manf* inactivation in *embryonic* development, rather pointed out Manf's putative developmental (maturation) role after birth. Reviewer 2 is right in writing that fine-grained development of stereocilia could be hampered in the mutant mice. This development could include the formation of stereocilia links required for hair bundle cohesion (lateral links) and mechanotransduction (tip links) and, thus, for the onset of hearing function. This topic is related to the criticism by reviewer 3 "the ms lacks physiological insight how Manf potentiates the effect of the mutant Cdh23 allele, because Cdh23 is a component of these links".

This is the *main criticism raised by the reviewers. We think that it is feasible to address the criticism* by new experiments. Therefore, we ask you to reconsider your decision. In the revised ms, we would provide evidence that stereocilia links are defective in *Manf* mutant mice under the C57BL6 background. Cdh23 is known to be a component of the stereocilia links and the C57BL6 background is known to carry the mutant *Cdh23* allele. Thus, we think that "adding insult to insult", i.e. ER stress due to *Manf* inactivation combined with ER stress due to *Cdh23* mutation (Blanco-Sánchez et al. 2014), impairs the development or maintenance of the stereocilia links, causing early-onset hearing loss. Along these lines, we can address the other criticism by reviewer 2 "the time course and frequency specificity of their stereociliary phenotype (splaying and/or fusion) does not match with the timing of the hearing phenotype". We would provide evidence of a tonotopic (high-to-low-frequency) gradient along the cochlea in the presence of the stereocilia links, a fact that could explain the prominent elevation of hearing thresholds specifically in the high frequencies and already at the onset of hearing. The argument that *Manf* KO mice have not been bred for enough numbers of generations to make it a pure-bred is not relevant for the data presented because we confirmed by sequencing the status of the *Cdh23* allele, i.e. we know that the animals analyzed are homozygous for the mutant allele.

The reviewers are inner ear experts with excellent comments. Reviewer 1: "Multiple mechanisms are associated with normal function of the inner ear, and pathogenesis leading to deafness. One of these, which is understudied, is the role of the endoplasmic reticulum (ER) in hearing and ER stress associated with deafness. Given the sensitivity of hair bundles and ribbon synapses to stress, and the ubiquity of ER stress in many diseases' pathogenesis, this is a particular compelling area to study. The work provides an advance in this area, outlining in detail the effects of ER stress and contribution to hair cell damage and the pathophysiology of deafness". The other reviewer commented that "the ms is technically strong". Based on these supportive comments and the fact that the criticism raised is not extensive and is possible to address, we cordially ask for reconsideration allowing us to revise the ms.

We strongly believe on the impact of our data to the auditory field and more broadly to the cell biological research on ER stress, and we think that your journal would be an appropriate forum for these data. We are looking forward to hearing from you soon.

Yours sincerely,

Ulla Pirvola

Corresponding author

Subject: *Re: Life Science Alliance Manuscript - Editorial Decision
LSA-2021-01068-T

Dear Dr. Pirvola,

We are pleased to share with you that if you are able to address the Editor comment 2 and 3 with additional data, as you have laid out in your email below, we would be happy to re-consider the manuscript at LSA.

We would advise you to submit an appeal with a pbp rebuttal and a revised manuscript (that includes the additional data) when you are ready.

Please let us know if you have any further questions.

Best,
Shachi

Life Science Alliance manuscript #LSA-2021-01068-T

REBUTTAL TO THE COMMENTS RAISED BY THE REFEREES

We thank the editors and the referees for the many constructive comments and suggestions. We have carried out extensive new experimentation to address the concerns and we have made necessary additions and changes to the text and figure material.

Editor #1, The advance of this manuscript over your previous publication, Herranen et al, 2020, remains very limited. While confirmatory studies are acceptable at LSA, we agree with the reviewer that the two publications are very similar with a lot of overlap in data and takehome message.

Response #1: We will first address the criticism that the current ms is very similar with our previous publication (Herranen et al., 2020). This criticism was raised by one out of the three referees. In both cases, the same mutant mouse model (*Manf* cKO mice) has been studied to understand the importance of the maintenance of ER homeostasis in the cochlear cells and in hearing function. Should one consider only the final functional phenotype, the end result is the same: severe sensorineural hearing loss. In the previous paper, we showed that the hearing loss at adulthood was accompanied by ABR threshold elevations and robust hair cell loss and we did not conclude anything more precise about the underlying cellular events nor the temporal progression of the hearing loss. Now that we have continued this research, we found that the causes of the hearing loss are more fundamental and are initiated much earlier. This allowed us to delve deeper into the topic. To add, our finding of an early-onset hearing loss phenotype in a *MANF*-variant patient brings more interspecies relevance to our analysis of young *Manf*-inactivated hair cells.

In the current ms, we show that the disruption of the critically important functional domains of hair cells, the hair bundle and ribbon synapses, is the primary cause of hearing loss in *Manf* cKO mice. We show that the hair bundle impairment is present already when hearing function starts at the juvenile life. We deal with the mechanisms that drive the progressive hair bundle pathophysiology. These novel findings show how the ER stress related pathology affects cell function long before inducing activation of the pro-apoptotic pathways. We think that this is important for a broader audience; in most cases pathological ER stress is studied in the context of cell death, while the current manuscript show its importance in driving dysfunction of cells.

Here, we specify the novel findings of the current manuscript:

- (1) The OHC hair bundles show abnormal structure from the hearing onset, which corroborates the ABR results at this stage and present the plausible proximal cause of the DPOAE threshold elevations.
- (2) We demonstrate a novel molecular mechanism of how the fusion of stereocilia in the OHC hair bundle proceeds. Stereocilia fusion has been described previously in pathological contexts, but here, we are able to deepen the understanding of the underlying mechanisms.
- (3) We show that IHC synaptopathy has an early onset and that it involves the loss of both pre- and post-synaptic structures, indicating non-functionality of the synapse. We deepen the knowledge of the spatial pattern of ribbon synapse loss across the modiolar-pillar axis of IHCs, important since it is known that noise-induced synaptopathy has a spatial preference linked to the functional characteristics of the synapses. We add data on the ultrastructure of the ribbon synapse by showing morphological abnormalities in the synaptopathic IHCs.
- (4) We show that *Manf* inactivation does not abolish neuroplastin expression in OHC hair bundles. Recent data show that MANF can bind neuroplastin, and neuroplastin has been suggested to be the long-sought receptor for MANF. This is important considering other recent data that neuroplastin is

required for PMCA2 expression in OHC stereocilia. However, we show that the stereocilia fusion event leads to neuroplastin and PMCA2 downregulation and thereby degraded Ca^{2+} clearance.

(5) We show novel human audiological data by analysing a loss-of-function *MANF* patient.

In all, in the published paper we revealed the role of *MANF* as a survival factor. In the present paper, we demonstrate the important contribution of *Manf* (ER homeostasis) to the maintenance of hair cell physiology and function, independently of cell death.

Editor #2, Reviewer 2 points out that the *Manf*-KO phenotypes could be secondary to a defect in embryonic development in general, and that *Manf*-KO mice have not been bred for enough numbers of generations (at least 10 would be required) to make it a pure-bred

Response #2: In the original ms., we showed that all hair cells (P22) and ribbon synapses are formed (P22) in cKO cochleas. We have added to revised ms data on the OHC hair bundle status in the immature (P9) mutant mice. We do not see defects at the level of hair bundles, for example bundle abnormalities seen in Usher mutant mouse models. We have addressed the question of the contribution of developmental abnormalities to the altered bundle phenotype in our response below to referee#2 and dealt with this topic in the revised ms (results lines 121-127; discussion lines 350-363, revised ms Figs 3 and S2).

While we have not bred the CBA/Ca KO mouse line to a pure bred, we have confirmed that all individual analysed lacked the *ahl* variant of *Cadherin-23*. Thus, we think that the requirement of a pure bred is not relevant. The data are summarized in Table 1.

Editor #3, Since the paper is a mechanistic follow-up of the previous study, the point made by Reviewer 3 that all of this detail does not necessarily provide meaningful physiological insight into the mechanism by which *Manf* essentially is potentiating the already-known effect of the *Cdh23* *Ahl* allele is also of concern.

Response #3: We thoroughly agree with the comments of referee#3. They are essentially same as those by referee#2. We specify our conclusions in our responses below to referees#2 and #3. The topic has been discussed in revised ms (discussion lines 350-363).

Here we list the experiments made during the revision process to address this topic:

- (1) We studied the pre-hearing developmental aspects of OHC stereocilia to elucidate whether *Manf* deficiency could potentiate some early deficit that the *Cdh23^{ahl}* missense mutation alone does not trigger. We were able to find that the OHC hair bundles of *Manf* cKO B6 (P9) mice are structurally comparable to age-matched control B6 mice. We could find that both genotypes develop a complement of inter-stereociliary links, many of which are known to contain *Cdh23*. We could not directly show if and how *Manf* inactivation (ER stress) plays together with the *Cdh23^{ahl}* mutation. We would like to point out that very little is understood of the molecular/cell biological mechanism how *Cdh23^{ahl}* mutation itself leads to progressive hearing loss typical to the B6 mouse strain.
- (2) In our attempts to reveal the mechanisms of the early-onset hearing loss of *Manf* cKO mice, we studied the possibility that PMCA2-dependent Ca^{2+} clearance is impaired in OHC stereocilia. This was an attractive possibility since *Manf* has been recently show to be able to bind neuroplastin in cell culture. Another recent study nicely demonstrated the tight association between neuroplastin and PMCA2 in OHC stereocilia (both papers cited in revised ms). We did not find *Manf* to be indispensable for the interaction of neuroplastin with PMCA2. However, we show in revised ms that the end result of the OHC pathology in adult *Manf* cKO mice, stereocilia fusion, is associated with downregulation of neuroplastin and PMCA2 and, therefore, with impaired Ca^{2+} dynamics. While it is speculative to assume that stereocilia fusion proceeds always the same way (even without *Manf* inactivation or ER

stress), this can be useful knowledge when it comes to studying the mechanisms underlying the loss of OHC function or OHC death.

Reviewer #1 (Comments to the Authors (Required)):

Line 42.

We propose the involvement of this mechanism in the pathophysiology of Usher syndromes. There is no evidence that suggests this mechanism is involved in US, which is a combination of deafness and RP. The only phenotype described in the manuscript is deafness. Please remove and replace "Usher syndromes" with "deafness." While *Cdh23* is associated with Usher syndrome, there is no additional evidence regarding this syndrome, rather than deafness alone.

Response: This last sentence of the abstract has been omitted from the revised ms.

Line 189. Remove 'data not shown'. Either show the data in supplementary or take out this sentence. It is no longer acceptable to make this statement.

Response: These data are added to Fig S3 in revised ms.

I am unable to access Montaser et al paper without payment; therefore, it is hard to judge if the data on the human patient is novel, or had already been described in the previous manuscript.

Response: Thus far, only two human patients carrying *MANF* mutations have been found. Yavarna et al. dealt with Case 2 and shortly reported "deafness" in this patient. Montaser et al. dealt with Case 1 and Case 2 and shortly reported in their table that both patients suffer "bilateral sensorineural deafness". Because of this very limited description of the audiological status in the two cases, we have described in more detail the audiological profile of Case 1, thanks to the high-quality audiological work by the Turkish Dr. K. Demir, a collaborator in our paper. This has been more clearly described in the revised ms (results lines 298-301; discussion lines 445-451).

Reviewer #2 (Comments to the Authors (Required)):

1. The previous publication from the lab is very similar to this study with a lot of overlap. I understand that the mice have been investigated at an earlier stage but I think the main messages of the papers remain very similar. The study is fairly descriptive and does not offer any mechanistic advance on what is already established

Response: We disagree with this comment, but have taken it very seriously. It is our major mistake that we did not clearly express the novel messages in original ms. Above as a response to Editor's comment#1, we specify in detail the new findings of our ms, including the novel data in revised ms.

2. What cells does the *pax2* cre target? There is no explanation or characterisation of *manf* expression or lack of it.

Response: *Pax2* is expressed from the early-embryonic stages onward in the inner ear; in the otic epithelial cells and in spiral and vestibular ganglion sensory neurons (Ohyama and Groves, 2004). We have added this missing information (materials and methods lines 669-672). Thanks! In the Herranen et al. paper, we showed the expression pattern of *Manf* in the cochlea (immunostainings and LacZ expression).

3. Are the effects of *manf* KO really related to a direct effect of *manf* or is it a developmental indirect effect? For instance, could it be something developmental related to mutation in *cdh23*? This needs to at least be clearly explained in the ms i.e. that all the effects could be secondary to some developmental defect.

Response: Please see above the response to Editor#2 comment. This referee addressed in essence similar questions as referee#3. Revised ms includes a description of the P9 OHC hair bundle status of mutant mice, analysed by SEM (results lines 121-127; revised ms Figs 3 and S2). The bundle morphology at this immature stage appears indistinguishable from controls and the disarray seen at the onset of hearing (P15) is not yet evident. We cannot exclude abnormalities related to development. We have discussed this in revised ms (lines 350-363):

Revised ms, discussion: Manf cKO B6 mice displayed OHC hair bundle disarray already at juvenile ages (P15, P22). The bundle disarray at this stage of onset of hearing was subtle. However, DPOAE and ABR thresholds were already prominently raised. Even though we did not see bundle abnormalities at the immature stage (P9), we cannot exclude possible fine-grained abnormalities related to development that could contribute to the hearing impairment. The mechanisms are elusive. Already in immature OHCs, Manf inactivation could perturb the production or trafficking of stereociliary proteins, particularly the mutant Cdh23 protein (B6 background). These perturbations could affect the structure and function of tip links and transient lateral links, manifested at the onset of hearing as problems in hair bundle cohesion and hair cell dysfunction. The OHC pathology in Manf cKO mice progressed thereafter. Chronic ER stress in older OHCs might exacerbate the effects of the Cdh23^{753G→A} mutation, known to drive age-related hearing loss in B6 mice. However, giving definitive proof for this concept in the adult mouse cochlea in vivo would require novel methods to measure the ER stress dynamics in hair cells. Additionally, it would be very important to show that this Cdh23 missense mutation indeed elicits ER stress in cochlear hair cells similarly as shown for the Cdh23 mutation in the neuromast hair cells of the sputnik zebrafish model (Blanco-Sanchez et al., 2014).

4. The mouse genetic background of the KO seems quite a mess. Five or six generations are not enough to get pure breed, at least 10 generations are required. This should be explicitly explained. Comment: (Also response Editor#2 comment). To be able to perform this study in a decent time constraint (not waiting for a few years to get the pure bred *Manf* KO CBA line), we have sequenced all our CBA individuals for the *Cdh23* gene (wild type/mutant heterozygous/mutant homozygous). Table 1 summarized these data.

5. The controls should also be cre positive with non-mutant *manf*. This would ensure that the *Pax2* cre expression is not having any direct effect.

Comment: As controls, we have used wild types, *fl/+* and *Cre+* mice. We have now changed this throughout the revised ms, so that we compare mutants to controls and we define the *control* status (materials and methods, line 672). Just to mention that we used in a recent paper the same Cre-line for conditional gene (*Mdm2*) inactivation in the inner ear (Laos et al. Sci Rep 2017).

6. Have the afferents and ribbon synapses been characterised in the OHCs? This would be a good comparison to make with the IHCs. Are the mechanisms affecting the synapses in both hair cell types similar or not?

Comment: In revised ms, we show a comparable OHC ribbon status in *Manf* cKO and control mice (revised ms Fig. S5, CtBP2 data). We have also studied these synapses using TEM, but the amount of OHC ribbons is too small to put it to a publication. This is an interesting topic. We do not have any answer to the IHC-OHC ribbon synapse difference. Is the ER differently located in the basal pole of IHCs and OHCs? Are there differences in Ca²⁺ buffering capacity at these locations, considering that ER stress is linked with ER-Ca²⁺ depletion? Etc. This is too speculative to add to the discussion.

7. The information on the human study is very limited. There is nothing mentioned of any other underlying defect or mutation that could bring the *manf* mutation into effect. The author's show that the *manf* KO has no affect by itself so what is going on in this patient?

Response: We have added to revised ms that there is no information available of any other underlying mutations contributing to the patient's hearing loss. Parents were heterozygous for *MANF* variants

(Montaser et al., 2021). No information is available of their hearing status (results lines 298-301). We have extended this discussion in the revised ms (discussion lines 447-451). As we say, there is a future need to screen for *MANF* variants in the hearing loss databases, without the possible diabetes bias, which is the case with the few *MANF* patients thus far found.

8. The IHC controls in figure 3 should be shown since the expression patterns seem to be different to the OHCs. This should also be explained.

Response: These data are shown in Fig. S3 in revised ms.

9. In figure 5N why is only one mouse used? Why not use both mice for this? The numbers of mice are very low anyway at $n=2$ for the analysis of TEM ribbon morphology so at least use all you have. Were the ribbons in CD-1 mice investigated at all? It would be interesting to see if the numbers are comparably low, although I understand there is no control for these.

Response: We agree with the referee about the need for higher sample size. In the revision process, we added 2 cochleas of both genotype to the analysis (now $n = 3$ in both cases). We could not confirm the original finding of increased proportion of double ribbons in IHCs of mutant cochleas. The variability was too high for statistical power with three specimens. Therefore, we have omitted the double ribbon data from the revised ms. We would like to point out that the amount of thin TEM sections analyzed per cochlea is measured in the hundreds in our work.

We are currently investigating ribbon synapses of CD-1 mice in another project, by counting CtBP2-positive ribbons. We have not done TEM analysis with CD-1 mice. To briefly summarize, CD-1 wildtype mice suffer early-onset, severe cochlear synaptopathy.

10. If you are drawing conclusions from data then they should be shown - line 173-179

Response: These data are shown in Fig S3 in revised ms.

11. The figures should be labelled in a way that would make them easier to read. Write what the colors of the staining represent, label what is in the picture i.e. WT or cKO etc. Also label the axes on the human hearing data.

Response: Thanks for this good comment that has brought clarity to our figure plates. We have worked on this in all figure plates and have added the missing axes labels. For example, the original Fig 3 is thoroughly modified (= revised ms Fig 5).

Reviewer #3 (Comments to the Authors (Required)):

...one critical timing problem limits the impact of their conclusion, Specifically, they show that *Manf* cKO mice have significant hearing loss, particularly at high frequencies, as early as P15 (shortly after the onset of hearing). In fact, the high-frequency thresholds are largely the same at P15 and P56. However, they find evidence of subtle stereociliary splaying in both high- and low-frequency outer hair cells at P22 (though they do not show the images for the low-frequency OHCs), and only evidence of stereociliary fusion at P56. Therefore, the time course and frequency specificity of their stereociliary phenotype (splaying and/or fusion) does not match with the timing of the hearing phenotype. This calls into question any conclusion they may draw into the specific mechanistic role through which *Manf* deficiency causes hearing loss through structural changes in the stereocilia, which is a central point of the paper.

Response: This is an astute observation and the question is important as hearing function was impaired in the mentioned young mice and, correspondingly, in the human patient. We have approached the question in revised manuscript with both new experiments and discussion. This referee addressed in essence similar questions as referee#2. Please see above the response to Editor#2 comment and the response to referee#2.

In addition to the information delivered in those responses:

- (1) We have added to revised ms images of the low-frequency OHC stereociliary splaying at the juvenile stage and at adulthood (revised ms Figs S1, 4B and F). We have gathered all data on hearing measurements to the same figure (revised ms Fig 1), allowing the reader to follow hearing deterioration in time.
- (2) In the paper by Li et al. (cited in introduction of revised ms), the phenotype of *Tmtc4* KO mice was studied. *Tmtc4* is a regulator of ER-Ca²⁺ dynamics. It is functioning at the level of SERCA pumps. *Tmtc4* KO B6 mice were crossed onto FVB/NJ (FVB) background. The altered phenotype of *Manf* cKO B6 mice depends on B6 background, while this was not the case with *Tmtc4* KO FVB mice. This difference is most likely due to different mechanisms of these molecules in the ER. According to a recent study led by a co-author in our ms (Kovaleva et al. bioRxiv repository), MANF can bind IRE1 α and regulate its activity. IRE1 α is the proximal component in a central UPR branch that regulates both adaptive and pro-apoptotic UPR. Hence, as an intimate partner of the UPR machinery, *Manf* could regulate the production and trafficking of mutant *Cdh23* (also taken into account the data in Blanco-Sánchez et al., 2014 paper). What we understand and what is written in the Li et al. paper, details how *Tmtc4* as a regulator of ER-Ca²⁺ dynamics converges on the UPR are not so clear. However, differences in these mechanisms might explain the requirement of the B6 background (including the *Cdh23 ahl* mutation) in *Manf* inactivation only. We would like to point out that the biochemical assays in the Li et al. paper were made on whole cochleas. It tells very little of the molecular mechanisms in hair cells.

November 1, 2021

RE: Life Science Alliance Manuscript #LSA-2021-01068-TR-A

Dr. Ulla Pirvola
University of Helsinki
Molecular and Integrative Biosciences Research Programme
Viikinkaari 1
Helsinki 00014
Finland

Dear Dr. Pirvola,

Thank you for submitting your revised manuscript entitled "MANF, an ER homeostasis regulator, promotes stereocilia and synapse integrity in auditory hair cells". We would be happy to publish your paper in Life Science Alliance pending final revisions necessary to meet our formatting guidelines. Please address Reviewer 2's remaining points in this final revision.

- please upload your main manuscript text as an editable doc file
- please upload your Table in editable .doc or excel format
- please upload your main and supplementary figures as single files
- please consult our manuscript preparation guidelines <https://www.life-science-alliance.org/manuscript-prep> and make sure your manuscript sections are in the correct order
- please use the [10 author names, et al.] format in your references (i.e. limit the author names to the first 10)
- please add an Author Contributions section to your main manuscript text
- please add your main, supplementary figure, and table legends to the main manuscript text after the references section
- please add callouts for Figures 3A-F; 4I, 6A-C to your main manuscript text

A. FINAL FILES:

B. MANUSCRIPT ORGANIZATION AND FORMATTING:

Sincerely,

Reviewer #1 (Comments to the Authors (Required)):

The authors have fulfilled the requests made by the reviewers. Given this is my 2nd review, I will not repeat my summary from the previous review. No additional issues need to be addressed.

Reviewer #2 (Comments to the Authors (Required)):

MANF, an ER homeostasis regulator, promotes stereocilia and synapse integrity in auditory hair cells

Summary

This study by Ikäheimo et al investigates how endoplasmic reticulum (ER) stress elicits hair cell pathology, using mouse models with inactivation of Manf (Mesencephalic astrocyte-derived neurotrophic factor). The manuscript has been substantially revised and has addressed most of my previous concerns. I just have a few remaining questions.

1. The effect of the manf conditional knockout in the cochlea is completely different between IHCs and OHCs even though both have similar molecules that should be regulated in similar ways. The stereocilia are affected only in the OHCs and the ribbon synapses only in the IHCs when both cell types have these structures. This should be made clearer throughout the manuscript. The generalised comment that hair cells have abnormal hair bundles and ribbon synapses does not apply to all and should be changed - even in the title. This finding is also very interesting and shows that the ER stress, Ca²⁺ dynamics, or cell metabolism are completely different between the two hair cell types.

2. You talk about finding the mechanisms behind the bundle fusion in the introduction and discussion. I don't think you have a mechanism since there is no indication how a lack of manf affects the bundle protein expression. You have some indication of the different molecules affected but not why or how they are affected.

3. You should show the hair bundles of the IHCs wherever they have been cropped out of the pictures. It is an important point that the bundles are normal compared to the OHCs. This also includes the immunostaining images - the IHCs should be shown and not put in the supplementary information as it is an important difference. You should also show that the cuticular plate phalloidin intensity is normal in cKO IHCs and that neuroplastin and PMCA2 expression is normal in IHCs.

4. The reduced PMCA2 expression in OHCs could be a consequence of lack of MET current and so the expression goes down simply because it is not required. This could be mentioned in the discussion.

5. How do you know that it is specifically the Cdh23 mutation that is the important factor in the 6J mice? You remove the manf phenotype when you cross with a CBA mouse, but that could be due to the background of the CBA mouse and not just to the Cdh mutation. Ideally this would have been done using a repaired version of the same mouse background. For this reason you should be careful when you relate your findings specifically to the Cdh23 mutation. I realise it would not be feasible to repeat the experiments.
6. For the IHC synapses it would be useful to have a count of the homer spots per IHC i.e. not just the percent of colocalised spots. This would show whether there are still the same number of afferent contacts per cell or whether they are also lost.
7. If you say that the IHC ribbons are abnormally large in the cKO they should be quantified and tested properly. Can you measure their width or surface area for example?
8. Without being a clinical audiology expert the results from the human study are hard to understand from the limited description given. The ABR waveforms are not clear and are impossible to read or make sense of without a normal control condition. Why bone conduction was used and what does this show? Why masked and unmasked and what does this mean? From the figures it looks like hearing is improving with age not declining.
9. Results line 108 - not "all hair cells" are shown in Fig 2 - but this should be changed according to point 3 above to include IHCs.
10. Line 146 - how do you know there is an "apparent lack of stereocilia deflection"? For instance how stiff is the fused bundle? Just remove the comment.
11. Line 228 - there is a "data not shown" - It should be shown if it is referred to.

Reviewer #3 (Comments to the Authors (Required)):

Thank you for your revisions. They do make the manuscript stronger. The mechanisms by which Manf deficiency potentiates Cdh23^{ahl} hearing loss are clearly complex and require further investigation, but the authors do a good job at providing technically sound and detailed description of the stereociliary changes in this model that will be of value to further understanding of the role of ER stress in stereociliary homeostasis/maintenance and progressive hearing loss.

November 8, 2021

RE: Life Science Alliance Manuscript #LSA-2021-01068-TRR

Dr. Ulla Pirvola
University of Helsinki
Molecular and Integrative Biosciences Research Programme
Viikinkaari 1
Helsinki 00014
Finland

Dear Dr. Pirvola,

Thank you for submitting your Research Article entitled "MANF supports the inner hair cell synapse and the outer hair cell stereocilia bundle in the cochlea". It is a pleasure to let you know that your manuscript is now accepted for publication in Life Science Alliance. Congratulations on this interesting work.

DISTRIBUTION OF MATERIALS:

Again, congratulations on a very nice paper. I hope you found the review process to be constructive and are pleased with how the manuscript was handled editorially. We look forward to future exciting submissions from your lab.

Sincerely,
